# Statistical learning attenuates visual activity only for attended stimuli

**David Richter\*, Floris P de Lange**

Donders Institute for Brain, Cognition and Behaviour, Radboud University Nijmegen, Nijmegen, Netherlands

**Abstract** Perception and behavior can be guided by predictions, which are often based on learned statistical regularities. Neural responses to expected stimuli are frequently found to be attenuated after statistical learning. However, whether this sensory attenuation following statistical learning occurs automatically or depends on attention remains unknown. In the present fMRI study, we exposed human volunteers to sequentially presented object stimuli, in which the first object predicted the identity of the second object. We observed a reliable attenuation of neural activity for expected compared to unexpected stimuli in the ventral visual stream. Crucially, this sensory attenuation was only apparent when stimuli were attended, and vanished when attention was directed away from the predictable objects. These results put important constraints on neurocomputational theories that cast perception as a process of probabilistic integration of prior knowledge and sensory information.

DOI: https://doi.org/10.7554/eLife.47869.001

**\*For correspondence:**
d.richter@donders.ru.nl

## Introduction

Previous experience constitutes a valuable source of information to guide perception and behavior. Extracting statistical regularities from past input in the environment to form expectations about the future has been shown to improve behavior in myriad ways (*Bertels et al., 2012*; *Hunt and Aslin, 2001*; *Kim et al., 2009*). Indeed, the acquisition of statistical regularities is thought to occur automatically (*Turk-Browne et al., 2009*) and affects behavior even in the absence of an intention to learn, or an awareness of, the regularities (*Fiser and Aslin, 2002*; *Brady and Oliva, 2008*). Given the significant behavioral and perceptual relevance of expectations, it is perhaps not surprising that the brain shows a remarkable sensitivity to statistical regularities. Many studies documented attenuated neural responses for expected compared to unexpected object stimuli in ventral visual regions subserving object recognition, both in terms of single unit spiking activity in monkeys (*Meyer and Olson, 2011*; *Kaposvari et al., 2018*) and in terms of non-invasively measured BOLD activity in humans (*den Ouden et al., 2010*; *Egner et al., 2010*; *Richter et al., 2018*; for a review see *de Lange et al., 2018*). This reduced response to expected stimuli has frequently been interpreted, within a predictive processing framework (*Friston, 2005*; *Rao, 2005*; *Rao and Ballard, 1999*), as signifying a reduction of prediction errors elicited by the stimulus when sensory input matches prior expectations. However, it remains largely unknown whether this sensory attenuation process to predicted visual stimuli is automatic, as its relation to statistical learning may suggest, or only apparent when the predictable stimuli are attended.

Indeed, research on visual statistical learning in monkeys has typically not manipulated attention, but only required monkeys to passively fixate in order to obtain reward (*Meyer and Olson, 2011*; *Kaposvari et al., 2018*), thereby precluding conclusions pertaining to the dependence of these predictive processes on attention. Many studies in humans, providing evidence for suppressed responses to expected stimuli, did require participants to attend the predictable stimuli (e.g., *den Ouden et al., 2010*; *Egner et al., 2010*; *Kok et al., 2012a*; *Richter et al., 2018*). On the other

hand, *den Ouden et al. (2009)* demonstrated attenuated responses to task-irrelevant expected stimuli, suggesting the possibility that the sensory consequences of statistical learning may not depend on attention. Similarly, *Kok et al. (2012a)* showed that the sensory attenuation for grating stimuli with an expected orientation was independent of whether the orientation feature was attended or not. Importantly however, in both these studies the expected or unexpected stimulus was the only stimulus presented on the screen, so even though the stimuli were not relevant, attention was not effectively disengaged by other stimuli. Without competition, it is likely that even a task-irrelevant stimulus will receive some attention.

Thus, at present it remains unclear whether statistical learning automatically results in altered neural responses to expected compared to unexpected visual stimuli, or whether this process hinges on the stimuli being attended. In order to answer this question, we exposed participants to sequentially presented pairs of object images. The first image predicted the identity of the second image, thereby making an image expected depending on temporal context. We recorded responses to expected and unexpected object images using whole-brain fMRI while participants performed one of two tasks. Either participants categorized the predictable, second object image as (non-)electronic (rendering the object images attended), or they classified a concurrently shown character (letter or symbol), presented within the fixation dot, as (non-)letter (rendering the object images unattended).

In brief, our results demonstrate strong sensory attenuation for expected object images within the ventral visual stream. Crucially however, expectation suppression was only evident when objects were attended and vanished when participants attended the concurrently presented alphanumeric characters at fixation. This suggests that sensory attenuation induced by statistical learning is not the result of an automatic integration of prior knowledge with incoming information, but hinges on attention, thus constraining neurocomputational theories of perceptual inference.

## Results

We exposed participants to statistical regularities by presenting object image pairs in which the leading image predicted the identity of the trailing image. During a learning session, participants performed a detection task of unpredictable upside-down images. On the next day, in the MRI scanner, participants were shown the same object image pairs, however unexpected trailing images were also presented; that is, images which were predicted by a different leading image. Crucially, participants either classified the trailing object as (non-)electronic, thus actively attending the predictable object, or classified a concurrently presented, but unpredictable, trailing character as (non-)letter, thus not attending the predictable object.

### Attention is a prerequisite for perceptual expectations

First, we investigated whether the sensory attenuation for expected object stimuli was equally present when participants attended the objects or not, focusing on our a priori defined ROIs (see *Figure 1A*): primary visual cortex (V1), object-selective lateral occipital complex (LOC), and temporal occipital fusiform cortex (TOFC). In all three regions, expectation suppression was robustly present when participants attended the objects (V1: $t_{(33)}$ = 3.573, p=0.001, $d_z$ = 0.613; LOC: $t_{(33)}$ = 3.860, p=5.0e-4, $d_z$ = 0.662; TOFC: $t_{(33)}$ = 5.133, p=1.2e-5, $d_z$ = 0.880), but absent when participants attended the characters at fixation; that is, when the predictable objects were unattended (V1: $t_{(33)}$ = −0.216, p=0.830, $d_z$ = −0.037; LOC: $t_{(33)}$ = −0.831, p=0.412, $d_z$ = −0.143; TOFC: $t_{(33)}$ = 0.072, p=0.943, $d_z$ = 0.012). Indeed, Bayesian analyses showed moderate support for the null hypothesis ($BF_{10}$ <1/3) of no expectation suppression in all three regions during the character categorization task (V1: $BF_{10}$ = 0.188; LOC: $BF_{10}$ = 0.253; TOFC: $BF_{10}$ = 0.184). The robustness of this distinct pattern of expectation suppression for the two conditions was statistically confirmed by an interaction analysis (expectation by attention interaction, V1:, $F_{(1,33)}$ = 7.706, p=0.009, $\eta^2$=0.189; LOC: $F_{(1,33)}$ = 12.580, p=0.001, $\eta^2$=0.276; TOFC: $F_{(1,33)}$ = 16.955, p=2.4e-4, $\eta^2$=0.339).

Thus, in V1, LOC, and TOFC, there was a significant suppression of BOLD responses for expected compared to unexpected object stimuli exclusively during the object categorization task. No such modulation of BOLD responses by expectation was observed in the objects unattended condition in any of the three a priori ROIs, and in fact, there was moderate evidence for the absence of such a modulation when objects were unattended. We repeated all ROI analyses within the same ROIs but

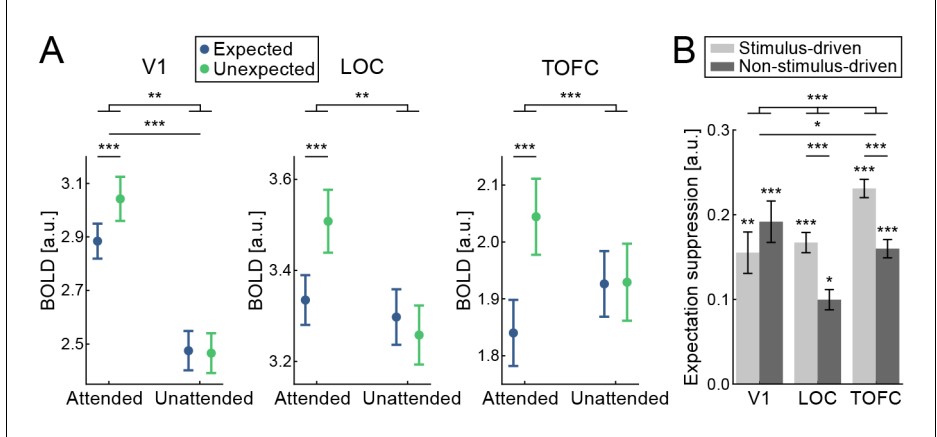

**Figure 1.** Expectation suppression within the ventral visual stream depends on attention. (**A**) Displayed are parameter estimates + /- within subject SE for responses to expected (blue) and unexpected (green) object stimuli during the objects attended task (attended) and objects unattended task (unattended). In all three ROIs, V1 (left), LOC (middle), and TOFC (right) BOLD responses were significantly suppressed in response to expected stimuli during the objects attended task. No difference was found between BOLD responses to expected and unexpected stimuli during the objects unattended task. The interaction effect between expectation and attention condition was significant in all three ROIs. (**B**) Expectation suppression in primary visual cortex is stimulus unspecific, and specific only in higher visual areas. Displayed is the average expectation suppression effect (BOLD responses, unexpected minus expected) split into stimulus-driven (light gray) and non-stimulus-driven (dark gray) gray matter voxels. Data are shown for the three ROIs, V1 (left bars), LOC (middle bars), and TOFC (right bars). Expectation suppression in LOC and TOFC was significantly larger for stimulus-driven than non-stimulus-driven voxels, while no such difference was evident in V1, indicating that expectation suppression in V1 was stimulus unspecific. Error bars indicate within-subject SE. Note, that the ROI masks in panel A and B differ, for details see: *ROI definition* and *Stimulus specificity analysis* in the Materials and methods section. *p<0.05. **p<0.01. ***p<0.001.

DOI: https://doi.org/10.7554/eLife.47869.002

The following source data is available for figure 1:

**Source data 1.** Expectation suppression within the ventral visual stream depends on attention.
DOI: https://doi.org/10.7554/eLife.47869.003

with different ROI sizes in order to ensure that our results were not dependent on the a priori but arbitrarily defined ROI mask size. Results were highly similar (i.e., the same effects showing statistically significant results) to those mentioned above within all three ROIs (V1, LOC, TOFC) for all tested ROI sizes, ranging from 100 to 400 voxels (800 mm$^3$ - 3200 mm$^3$) in steps of 100 voxels. Thus, our results do not depend on the exact ROI size but represent responses within the respective areas well.

We also examined how expectation modulated neural activity outside our predefined ROIs by performing a whole-brain analysis. Results of this whole brain analysis are illustrated in *Figure 2A*. The upper row in *Figure 2A* shows extensive clusters of expectation suppression throughout the ventral visual stream when objects were attended, but no difference when the objects were unattended (middle row), leading to a significant interaction (bottom row). These results complement our ROI-based analysis by showing that the observed expectation suppression effect is not unique to the a priori defined ROIs but evident throughout the ventral visual stream.

Outside the ventral visual stream, additional clusters of expectation suppression are evident in anterior insula and the frontal operculum, the precentral and inferior frontal gyrus, superior frontal gyrus and supplementary motor cortex, superior parietal lobule, as well as parts of the cerebellum. All significant clusters are summarized in a table in *Supplementary file 1*. Again, all these non-sensory clusters showed reduced activity for expected objects only when the object stimuli were

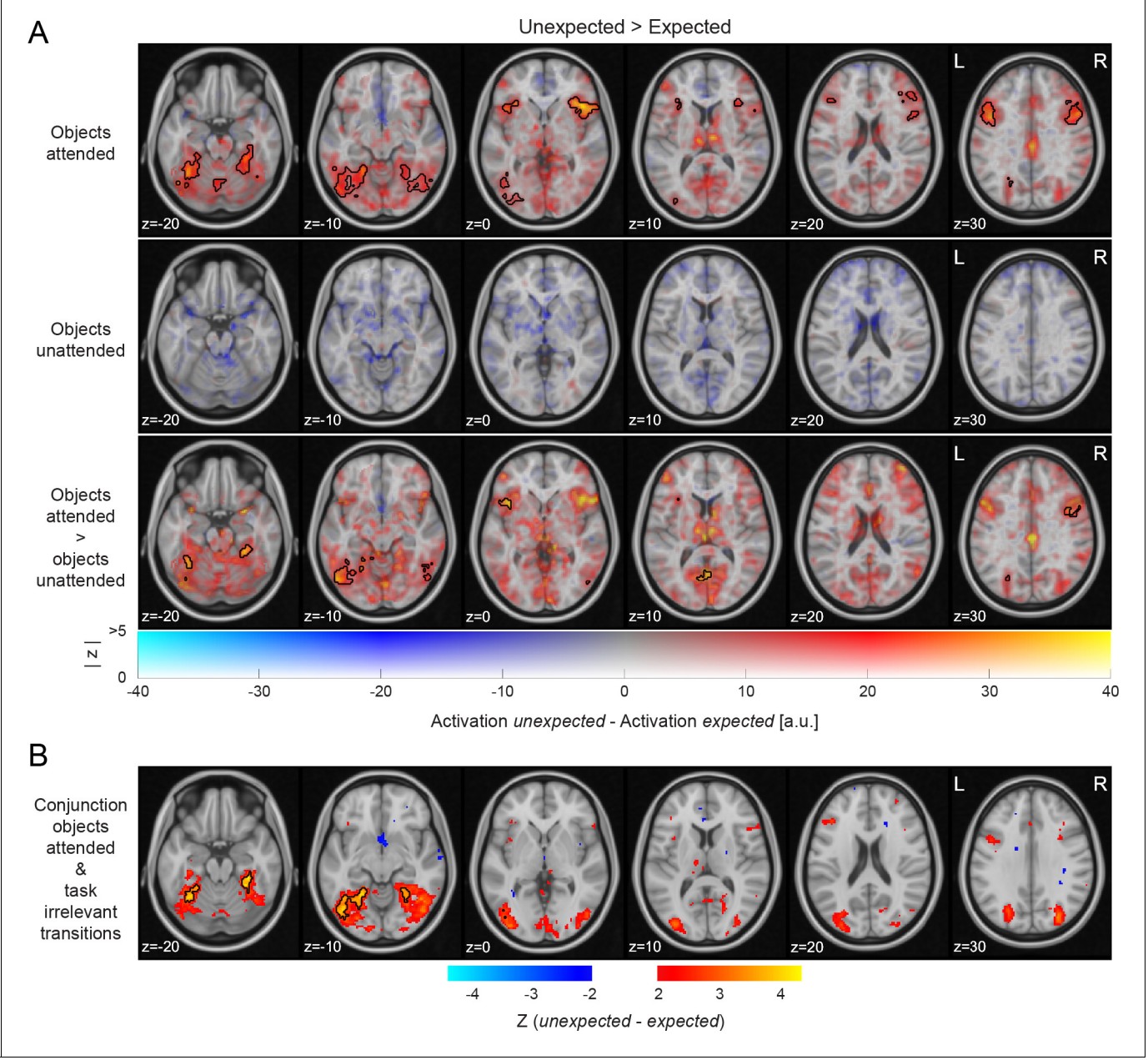

**Figure 2.** Expectation suppression across cortex for attended object stimuli only. (A) Widespread expectation suppression across cortex in the objects attended condition. Displayed are parameter estimates for unexpected minus expected image pairs overlaid onto the MNI152 2 mm template. Color indicates unthresholded parameter estimates: red-yellow clusters represent expectation suppression. Opacity represents the z statistics of the contrasts. Black contours outline statistically significant clusters (GRF cluster corrected). Significant clusters included major parts of the ventral visual stream (early visual cortex, LOC, TOFC), anterior insula, and inferior frontal gyrus during the objects attended condition (upper row). No significant clusters were evident in the objects unattended condition (middle row). The interaction (attended >unattended; bottom row) showed significant clusters similar to those of the attended condition, albeit less extensive. (B) Expectation suppression across the ventral visual stream for attended objects, but with task-irrelevant predictions. Displayed are z statistics of the contrast unexpected minus expected of the conjunction: *attended task-relevant predictions* $\bigcup$ *task-irrelevant predictions*; data of task-irrelevant predictions from *Richter et al. (2018)*. Exclusively the ventral visual stream clusters showed significant expectation suppression in this conjunction, while all non-sensory area clusters were no longer significant. Thus, only the ventral visual stream clusters displayed a sensitivity to conditional probabilities, irrespective of whether predictions were task-relevant or task-irrelevant, as long as the predictable stimuli were attended.

DOI: https://doi.org/10.7554/eLife.47869.004

The following source data is available for figure 2:

**Source data 1.** Expectation suppression across cortex for attended object stimuli only.

DOI: https://doi.org/10.7554/eLife.47869.005

attended and categorized. There was no significant modulation of activity by expectation anywhere in the whole brain analysis when the objects were unattended.

## Expectation suppression requires attention to the stimuli, but not their predictable relationship

During the object categorization task, the ability to form expectations about the trailing object stimulus was helpful for the participants, and indeed expected object stimuli were categorized more quickly and accurately (see Figure 5A and *Expectations facilitate object classification*). This begs the question whether the expectation suppression effect that we observed throughout multiple brain areas during the object categorization task reflects differences in task engagement. Participants had an incentive to (implicitly or explicitly) use their knowledge of the predictable relationship between the leading and trailing image to prepare their object categorization response. In order to examine which brain regions exhibited expectation suppression irrespective of the relevance of the predictable relationship between stimuli, we performed a conjunction analysis that highlighted regions that showed significant expectation suppression both in the current study (during the object categorization task) and in a similar study that we published previously (*Richter et al., 2018*). During this latter study, participants also attended the object stimuli, but were asked to press a button whenever an object appeared that was flipped upside-down. Upside-down images occurred rarely, and importantly, were not related to the (implicitly learned) statistical regularities. *Figure 2B* shows the whole-brain results of this conjunction analysis. Significant, bilateral clusters of expectation suppression were evident throughout most of the ventral visual stream. However, none of the non-sensory clusters showed significant expectation suppression during both experiments. Thus, only in the ventral visual stream we found strong and robust evidence for expectation suppression, regardless of whether the predictable relationship was task-relevant or task-irrelevant, as long as the predictable object pairs were attended.

## Stimulus specificity of the neural modulation by expectation

Next, we investigated the stimulus specificity of expectation suppression. Stimulus specificity concerns the question whether only stimulus-driven voxels or also voxels that were not (strongly) driven by the object stimuli displayed expectation suppression. The rationale was that an unspecific suppression effect (i.e., expectation suppression that is also evident in not stimulus-driven voxels) may result from global non-sensory effects, such as changes in general arousal or global surprise signals. On the other hand, stimulus-specific suppression effects, being limited to stimulus-driven voxels, are rather suggestive of a more specific suppression mechanism that selectively operates on the neural populations that represent the expected stimulus; for example, the dampening of stimulus-specific prediction errors as a result of a match between prediction and input.

All three ROIs were split into two populations of gray matter voxels, according to their stimulus responsiveness (stimulus-driven: responding to the object images; not stimulus-driven: not significantly responding to the object images), using independent data from the localizer run. There were strong differences between the ROIs in terms of the stimulus specificity of expectation suppression (*Figure 1B*; ROI x drive interaction: $F_{(1.245, 41.080)}$ = 7.651, p=0.005, $\eta^2$=0.188). Whereas there was clear evidence for a larger expectation suppression effect in stimulus-driven than not stimulus-driven voxels in higher visual areas (LOC: $t_{(33)}$ = 3.991, p=3.4e-4, $d_z$ = 0.684; TOFC: $t_{(33)}$ = 4.654, p=5.1e-5, $d_z$ = 0.798), suppression was not significantly different between stimulus-driven and not stimulus-driven voxels in V1 ($t_{(33)}$ = −1.057, p=0.298, $d_z$ = −0.181). Indeed, a Bayesian analysis indicated moderate support for the absence of a difference between stimulus-driven and not stimulus-driven voxels in V1 ($BF_{10}$ = 0.307). Of note, all sub-populations in all three ROIs showed significant expectation suppression (all p<0.05), suggesting that there is a general suppression of activity for expected stimuli in visual cortex, irrespective of whether the visual cortical area is driven by the stimuli. However, in later visual cortical areas (LOC and TOFC) there was significantly more expectation suppression in neuronal subpopulations that were driven by the stimulus, implying a more selective suppression mechanism in these areas.

## Surprising stimuli elicit a larger pupil dilation

In view of the suggestion that a global, stimulus unspecific response modulation may partially account for expectation suppression, we performed an exploratory analysis to examine whether surprising stimuli were associated with a stronger pupil dilation in our task. Pupil responses have been with linked with changes in arousal (*Reimer et al., 2014*; *Vinck et al., 2015*), which in turn may account for the stimulus unspecific suppression component. Moreover, pupil dilation scales with surprise (*Damsma and van Rijn, 2017*; *Kloosterman et al., 2015*; *Preuschoff et al., 2011*). Thus, this account would predict enhanced pupil dilation to unexpected compared to expected stimuli when objects were attended.

There was indeed a larger pupil diameter for unexpected compared to expected trailing images during the objects attended task (*Figure 3*, left). This difference emerged gradually starting ~600 ms after the onset of the trailing object image, and was significant between 1.5–2.8 s, as assessed with a cluster permutation test ($p_{cluster} = 0.017$). When objects were unattended, no significant difference in pupil diameter was found between the expectation conditions, and in fact, no timepoint surpassed the cluster formation threshold (i.e., all timepoints $p>0.05$ uncorrected; *Figure 3*, right). However, the expectation induced difference in pupil diameter was not reliably different between attended and unattended stimuli ($p_{cluster} = 0.393$). Thus, the data showed that the pupil was

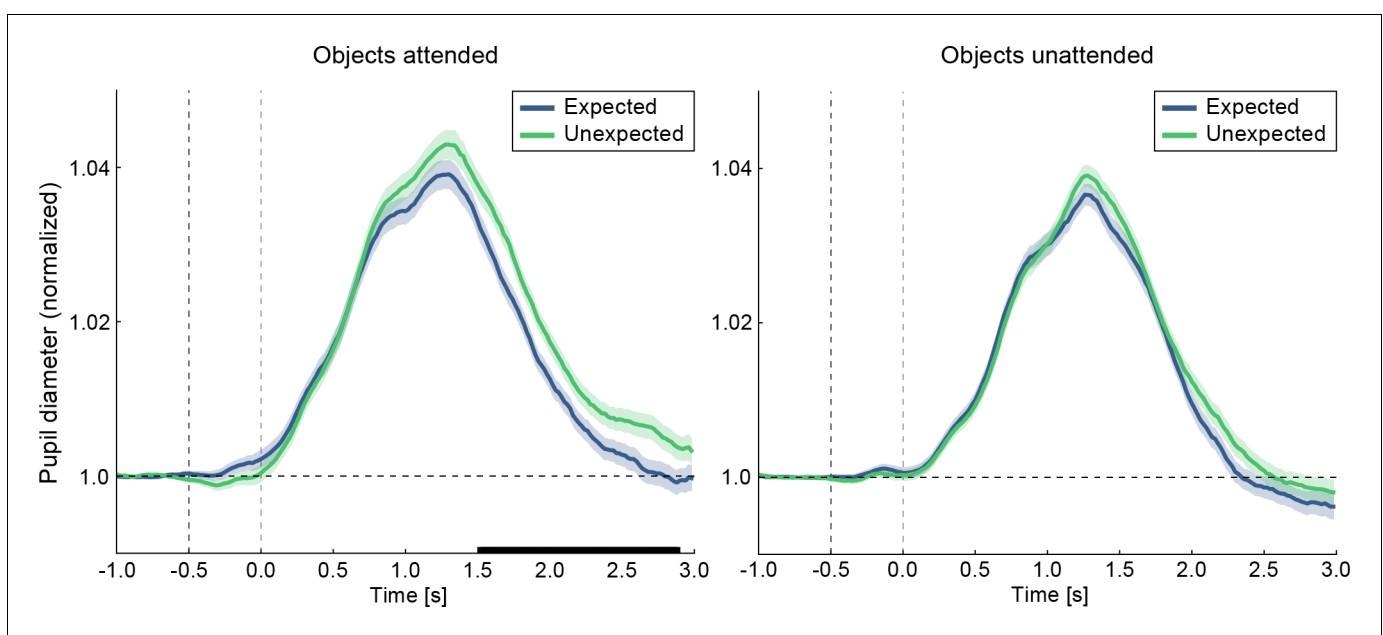

**Figure 3.** Larger pupil dilations in response to unexpected compared to expected stimuli during the objects attended task. Displayed are pupil diameter traces over time, relative to trailing image onset. Pupil diameter data for expected (blue) and unexpected (green) image pairs are shown for the objects attended task (left) and objects unattended task (right). The black line on the abscissa denotes statistically significant differences in pupil dilations between expected and unexpected images (cluster permutation test, p<0.05). In the objects attended condition significantly larger pupil dilations in response to unexpected images are evident between 1.52 to 2.88 s after trailing image onset (left). No significant difference is found in the objects unattended condition (right), nor in the interaction between conditions. The first vertical dashed line indicates leading image onset, the second vertical line trailing image onset. Shaded areas denote within-subject SE. Timepoints from −1.0 to −0.5 s served as baseline period.

DOI: https://doi.org/10.7554/eLife.47869.006

The following source data and figure supplements are available for figure 3:

**Source data 1.** Larger pupil dilations in response to unexpected compared to expected stimuli during the objects attended task.
DOI: https://doi.org/10.7554/eLife.47869.010

**Figure supplement 1.** Pupil dilation influences BOLD responses in V1.
DOI: https://doi.org/10.7554/eLife.47869.007

**Figure supplement 2.** Pupil dilation influences BOLD responses more in non-stimulus-driven than stimulus-driven V1 voxels.
DOI: https://doi.org/10.7554/eLife.47869.008

**Figure supplement 3.** No difference in baseline pupil size between attention tasks, nor expectation conditions.
DOI: https://doi.org/10.7554/eLife.47869.009

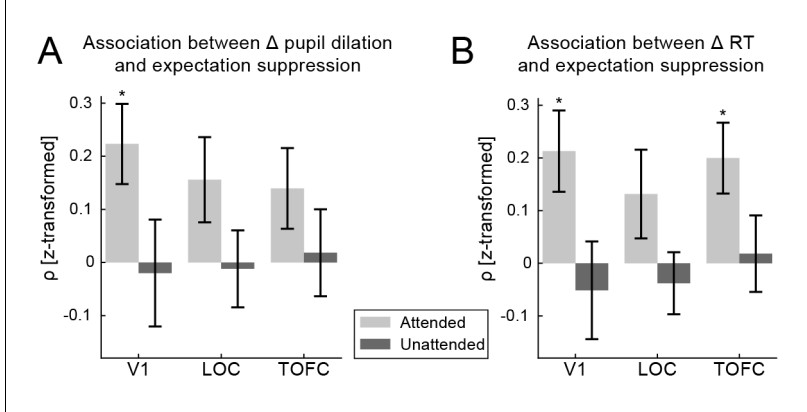

**Figure 4.** Expectation suppression is associated with pupil dilation differences and behavioral benefits of expectations. (**A**) Correlation of expectation suppression magnitude and pupil dilation differences due to expectation. When predictable objects are attended, trailing images that induce larger pupil dilation differences are also showing larger expectation suppression magnitudes in V1. No such association is evident when objects are unattended. (**B**) Correlation of expectation suppression magnitude and RT benefits due to expectation. When predictable objects are attended, larger RT benefits are associated with larger expectation suppression effects in V1 and TOFC. This association is absent when objects are unattended. Error bars indicate within-subject SEM. *p<0.05.

DOI: https://doi.org/10.7554/eLife.47869.011

The following source data is available for figure 4:

**Source data 1.** Neural effects of expectations are associated with pupil dilation differences and reaction time benefits.
DOI: https://doi.org/10.7554/eLife.47869.012

significantly more dilated for unexpected than expected objects when the images were attended, mirroring the results of the neural data – albeit, without a reliable difference between attended and unattended stimuli. This tentatively suggests that the enhanced BOLD responses to unexpected stimuli might be partially accounted for by a global mechanism, such as increased arousal in response to surprising stimuli.

## Expectation suppression and pupil dilations to surprising stimuli are associated

We explored whether expectation suppression and pupil dilation differences between unexpected and expected objects were associated. In other words, we sought for evidence of an association between the effect of expectations on pupil dilation and the expectation induced neural response attenuation. For this analysis we rank correlated expectation suppression magnitudes with pupil dilation differences for each participant. Results, displayed in *Figure 4A*, suggest that, when objects were attended, expectation suppression in V1 was more pronounced for trailing images that also resulted in larger pupil dilation differences ($t_{(31)}$ = 2.464, p=0.019, $d_z$ = 0.436). This association was not reliable in LOC ($t_{(31)}$ = 1.413, p=0.167, $d_z$ = 0.250; $BF_{10}$ = 0.466) or TOFC ($t_{(31)}$ = 1.401, p=0.171, $d_z$ = 0.248; $BF_{10}$ = 0.458). There was no correlation of pupil dilation differences and expectation suppression when stimuli were unattended in any of the ROIs (V1: $t_{(31)}$ = −0.159, p=0.875, $d_z$ = −0.028; $BF_{10}$ = 0.191; LOC: $t_{(31)}$ = −0.125, p=0.901, $d_z$ = −0.022; $BF_{10}$ = 0.190; TOFC: $t_{(31)}$ = 0.177, p=0.861, $d_z$ = 0.031; $BF_{10}$ = 0.192). There was no significant overall difference in the correlation strength between attended and unattended stimuli ($F_{(1,31)}$ = 1.892, p=0.179, $\eta^2$=0.058), nor between ROIs ($F_{(1.558,48.293)}$ = 0.134, p=0.823, $\eta^2$=0.004), nor their interaction ($F_{(2,62)}$ = 0.482, p=0.603, $\eta^2$=0.015). Thus, when stimuli were attended there was evidence for an association of pupil dilation and expectation suppression in V1.

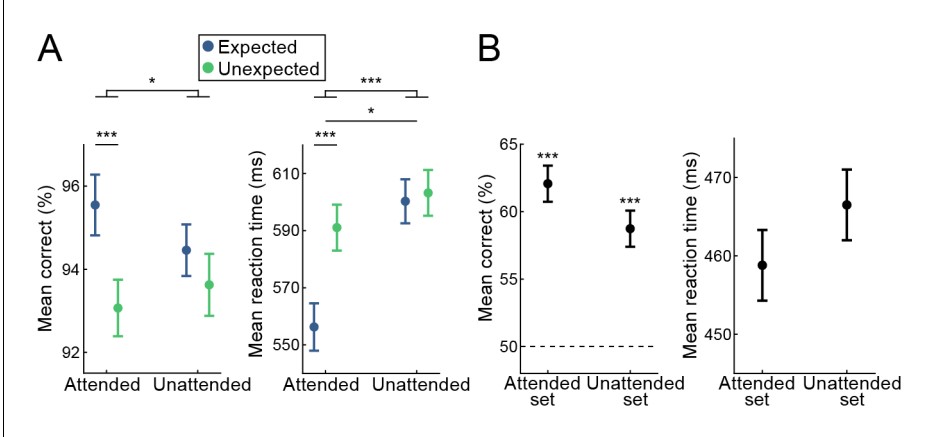

**Figure 5.** Behavioral results demonstrate statistical learning. (**A**) Behavioral benefits of expectations demonstrate statistical learning. Displayed are mean accuracy (left) and mean reaction time (right) + / - within subject SE. Responses to expected stimuli are significantly more accurate and faster, an effect exclusively observed during the objects attended condition. Thus, object identity expectations benefit behavioral performance during object classification and do not impact letter classification. (**B**) Pairs of both the objects attended image set and the objects unattended image set were classified significantly above chance, indicating a learning of the pairs for both conditions. Displayed are mean accuracy (left) and mean reaction time (right) during the post-scanning pair recognition task, + / - within subject SE. The dashed line indicates chance level. During the pair recognition task, no differences in either classification accuracy (left) or response speed (right) were observed between pairs previously belonging to the objects attended task compared to the objects unattended task. *p<0.05. ***p<0.001.
DOI: https://doi.org/10.7554/eLife.47869.013

The following source data is available for figure 5:

**Source data 1.** Behavioral results demonstrate statistical learning.
DOI: https://doi.org/10.7554/eLife.47869.014

## Expectations facilitate object classification

In order to assess whether, concurrent with the neural effects of expectations, behavioral benefits of expectations were evident, we analyzed behavioral responses during MRI scanning in terms of reaction times (RTs) and response accuracy. Overall, the objects attended (classify electronic items) and objects unattended task (classify characters at fixation) showed very similar response accuracies (attended: 94.3 ± 5.4% vs. unattended: 94.0 ± 6.6%, mean ± SD) and only minor differences in RTs (attended: 574 ± 150 ms vs. unattended: 602 ± 131 ms, mean ± SD). This supports the notion that both tasks were of approximately equal difficulty.

During the object categorization task, participants could benefit from the foreknowledge of the identity of the trailing object image, as they were asked to categorize the trailing image. Such a benefit would however not be expected during the character categorization task, as the participants could fully ignore the object stimuli during this task. This is precisely what we observed, both in terms of accuracy and RTs (*Figure 5A*). During the object categorization task, participants were more accurate ($W$ = 457, p=3.2e-4, $r_B$ = 0.536) and faster ($W$ = 9, p=3.8e-9, $r_B$ = −0.970) for expected compared to unexpected trailing object stimuli. Conversely, during the character categorization task, no such benefit was observed in terms of accuracy ($t_{(33)}$ = 1.600, p=0.119, $d_z$ = 0.274; $BF_{10}$ = 0.582) or RT ($W$ = 252, p=0.447, $r_B$ = −0.153; $BF_{10}$ = 0.273). The robustness of this distinct pattern of behavioral advantage for expected stimuli for the two conditions was statistically confirmed by an interaction analysis (accuracy: $F_{(1,33)}$ = 5.203, p=0.029, $\eta^2$=0.136; RT: $F_{(1,33)}$ = 37.543, p=6.6e-7, $\eta^2$=0.532).

## Neural and behavioral effects of expectations are associated

In order to explore whether the observed expectation suppression is associated with the behavioral benefits due to expectations, we correlated the magnitude of expectation suppression and the expectation induced RT benefits. Results, illustrated in *Figure 4B*, show that when the predictable objects were attended, behaviorally observed expectation RT benefits and neurally observed expectation suppression were associated in both, V1 ($t_{(33)}$ = 2.442, p=0.020, $d_z$ = 0.419) and TOFC ($t_{(33)}$ = 2.236, p=0.032, $d_z$ = 0.384), but no reliable correlation was found in LOC ($t_{(33)}$ = 1.384, p=0.176, $d_z$ = 0.237, $BF_{10}$ = 0.439). There was no association in any ROI when objects were unattended (V1: $t_{(33)}$ = −0.418, p=0.679, $d_z$ = −0.072, $BF_{10}$ = 0.199; LOC: $t_{(33)}$ = −0.374, p=0.711, $d_z$ = −0.064, $BF_{10}$ = 0.196; TOFC: $t_{(33)}$ = 0.179, p=0.859, $d_z$ = 0.031, $BF_{10}$ = 0.186). On average correlations were not reliably larger when objects were attended than when they were unattended (attention: $F_{(1,33)}$ = 2.920, p=0.097, $\eta^2$=0.081). The pattern of results was similar in all ROIs ($F_{(1.636,53.988)}$ = 0.615, p=0.513, $\eta^2$=0.018; interaction: $F_{(1.461,48.203)}$ = 0.381, p=0.619, $\eta^2$=0.011). Thus, there is some evidence that when the objects were attended, participants showed larger benefits (faster RTs) for expected trailing images for which they also showed larger magnitudes of expectation suppression in V1 and TOFC. These results suggest that the neural and behavioral effects of expectations are associated.

## No differences in association strength between attended and unattended object pairs

An alternative explanation for the absence of sensory attenuation for expected object stimuli during the character categorization task is that statistical regularities for the objects that are presented during this condition have simply not been learned. This explanation may be unlikely, because the vast majority of exposure to the expected pairs takes places in the learning session, during which the same task (upside-down image detection) was used for all image pairs. However, it is nonetheless important to ensure that statistical regularities were learned for the image pair sets of the object and the character categorization task. To empirically address this, we tested whether participants had explicit knowledge of the statistical regularities for all object pairs. During this post-scanning pair recognition task, participants were asked to indicate which one of two trailing images was more likely given the leading image. Participants indicated the correct trailing image with above chance accuracy for both, the set of object pairs that was previously attended (*Figure 5B*; performance = 62.1 ± 1.8%, mean ± SE; $t_{(33)}$ = 6.803, p=4.6e-8, $d_z$ = 1.167) and the set that was previously unattended (performance = 58.7 ± 2.2%; $t_{(33)}$ = 3.905, p=2.2e-4, $d_z$ = 0.670). There was no statistically significant difference in accuracy on the pair recognition task between these sets of objects ($W$ = 365, p=0.256, $r_B$ = 0.227; $BF_{10}$ = 0.737). Reaction times were also similar for both sets of objects (objects previously attended: RT = 458.8 ± 25.4 ms; objects previously unattended: RT = 466.5 ± 25.9 ms; $t_{(33)}$ = −1.208, p=0.236, $d_z$ = −0.207; $BF_{10}$ = 0.358). Thus, the image pairs belonging to both task conditions (objects attended and unattended tasks) were reliably learned, most likely during the extensive behavioral training session, and there was no evidence for a significant difference in the learning of associations for the two sets of object pairs. This strongly suggests that the differences in sensory attenuation between the two attention conditions are unlikely to be explained by differences in the strength of the association between the object pairs.

## Visual processing continues in the absence of attention

Finally, one may wonder whether the lack of expectation suppression when objects were unattended is due to the fact that object stimuli simply did not elicit strong activity in the ventral visual stream, as they were not in the focus of attention. Although all three ROIs showed reliable above-baseline activity also when objects were unattended (*Figure 1A*), and activity in LOC and TOFC was of similar amplitude during both conditions, the overall activity level may partly represent stimulus-unrelated activity. Therefore, in an explorative analysis, we assessed the strength of stimulus-specific activity in our three ROIs, by means of a decoding analysis of the trailing images. In brief, a multi-class decoder was trained to differentiate between the six trailing images per attention condition. The classifier was trained on data obtained in an independent localizer run, during which participants performed a separate task (detection of dimming of fixation dot). Performance of this decoder was tested on the mean parameter estimates per trailing image for each of the two attention conditions of the main

MRI task data. Because each task was comprised of six trailing images, chance performance was 16.7%. One-sample t-tests or Wilcoxon signed rank test (as applicable) showed that in each of the three ROIs (V1, LOC, TOFC) and tasks (objects attended, objects unattended) object identity could be decoded above chance (V1 attended: 81.1%; $W = 595$, p=3.3e-7, $r_B = 1$; V1 unattended: 84.8%; $W = 595$, p=3.2e-7, $r_B = 1$; LOC attended: 37.3%; $t_{(33)} = 6.303$, p=4.0e-7, $d_z = 1.08$; LOC unattended: 38.0%; $W = 583$, p=9.7e-7, $r_B = 0.96$; TOFC unattended: 25.0%; $W = 476$, p=0.002, $r_B = 0.60$), except in TOFC in the attended condition (TOFC attended: 19.6%; $W = 383$, p=0.143, $r_B = 0.287$; $BF_{10} = 0.388$).

Moreover, decoding accuracy was not different between the objects attended and unattended conditions in any of the ROIs (V1: $t_{(33)} = -1.197$, p=0.240, $d_z = -0.205$, $BF_{10} = 0.354$; LOC: $t_{(33)} = -0.214$, p=0.832, $d_z = -0.037$, $BF_{10} = 0.188$; TOFC: $t_{(33)} = -1.726$, p=0.094, $d_z = -0.296$, $BF_{10} = 0.697$). This suggests that the object stimuli evoked a reliable stimulus-specific activity pattern in all three sensory regions, which was not significantly different in strength between the two tasks (object categorization and character categorization). Note, the participants' task during the localizer run, which we used to train the classifier, was to detect a dimming of the fixation dot. As such, object stimuli were unattended during the localizer run, which may render the training data more similar in terms of attention allocation to the objects unattended task than the objects attended task. This may explain why decoding accuracy is similar, or even higher, for unattended compared to attended objects. More importantly, overall visual processing of the object stimuli was clearly present even when the objects stimuli were not attended, as the identity of the objects could be reliably decoded from neural activity patterns throughout the ventral visual stream when objects were unattended.

## Discussion

In the present study, we set out to investigate how sensory attenuation following visual statistical learning is modulated by attention. In line with previous studies (*Alink et al., 2010*; *den Ouden et al., 2010*; *Kok et al., 2012a*; *Richter et al., 2018*; *Summerfield et al., 2008*) we found a significant and wide-spread attenuation of neural responses to expected compared to unexpected stimuli. Crucially, we showed that attending to the predictable stimuli is a prerequisite for this expectation suppression effect to arise. While unattended objects led to reliable and stimulus-specific increases in neural activity, and object pairs were equally learned for these stimuli, there was no differential activity depending on whether the trailing object was expected or unexpected. Additionally, we found that higher visual areas exhibited stimulus-specific expectation suppression, whereas early visual cortex showed a global, stimulus unspecific suppression, possibly arising from a general increase in arousal in response to surprising stimuli.

### Attention is a prerequisite for expectation suppression

Our results show that a core neural signature of perceptual expectations, expectation suppression (*Alink et al., 2010*; *den Ouden et al., 2010*; *Kok et al., 2012a*; *Richter et al., 2018*), is only evident when attention is directed to the predictable object stimuli. Specifically, when participants engaged in an object categorization task, we found a wide-spread reduction of neural activity for expected compared to unexpected stimuli throughout the ventral visual stream (V1, LOC, TOFC), as well as several non-sensory areas (anterior insula, inferior frontal gyrus, precentral gyrus, and superior parietal lobule). Strikingly, no modulation of neural activity by expectation was found when attention was drawn away from the object stimuli.

Interestingly, by directly comparing our present data with a previous dataset, in which we used a similar design (reported in *Richter et al., 2018*), we established that expectation suppression is present throughout the ventral visual stream irrespective of whether predictions are task-irrelevant, as long as the object stimuli are attended. In contrast, the larger activity for surprising stimuli in non-sensory areas (insular, frontal and parietal cortex) was only observed in the context of task-relevant expectations. This suggests that neural activity in the ventral visual stream is modulated by conditional probabilities, as long as the stimuli are attended, while the modulations in non-sensory regions are probably reflecting differences in task demands, given that unexpected stimuli were more difficult to categorize (reflected by a cost in speed and accuracy). During the object classification task, unexpected objects may require response inhibition, reevaluation of the category, and thus a new

response decision. Given that the anterior insula has been associated with task control, action evaluation (*Brass and Haggard, 2010*), as well as general attentional processes (*Nelson et al., 2010*), and inferior frontal gyrus with response inhibition (*Aron et al., 2003*; *Aron et al., 2004*), the interpretation that the expectation modulation in non-sensory clusters may reflect task related aspects, but not conditional probabilities per se, appears well-supported by previous research.

Finally, our results also demonstrate that larger expectation suppression effects in V1 and TOFC are associated with increased reaction time benefits afforded by expectations when people are judging the predictable objects. This suggests that the observed expectation suppression effect may not merely constitute an epiphenomenon of more resource efficient neural processing. Instead, given the present data, it is plausible that the behavioral advantage of predicting stimuli may partially be rooted in improved and more effective sensory processing already at the early stages of visual processing. Predictions may thus help in converging more rapidly on an interpretation of the current sensory input, thereby contributing to faster reactions to expected than unexpected stimuli.

## No perceptual predictions without attention

Our results corroborate and extend earlier work by *Larsson and Smith (2012)*, who observed that stimulus expectation only affected repetition suppression when the stimuli were attended. However, they appear at odds with several previous studies that have reported expectation suppression in the visual system for stimuli that were not task-relevant and thus appeared unattended (*den Ouden et al., 2009*; *Kok et al., 2012a*; *Kok et al., 2012b*). However, in all these studies, while the predictable stimuli were task-irrelevant, attention was not effectively drawn away by a competing stimulus that required attention. While our attention manipulation is also based on task-relevance, we do engage attention elsewhere using a competing task. This is a crucial difference between the present and previous studies, because it is likely that any supraliminal stimulus, in the absence of competition, will be attended to some degree, even if it is not task-relevant, especially if the stimulus is surprising (*Horstmann and Herwig, 2015*). Indeed, synthesizing earlier and current findings, we can conclude that expectation suppression in the visual system occurs irrespective of exact task goals and relevance of the predictable objects and their predictable relationship, but it is abolished by drawing attention away from the stimuli. This suggests that the integration of prior knowledge and sensory input is gated by attention – that is, prior knowledge only exerts an influence on stimuli that are in the current focus of attention, instead of automatically and pre-attentively modulating sensory input as an obligatory component of perceptual processing.

It is however possible that other, more 'stubborn' prior expectations (*Yon et al., 2019*) that are derived over longer (ontogenetic or phylogenetic) time scales may persist even when attention is drawn away, such as perceptual fill-in during the Kanizsa illusion (*Kok et al., 2016*). Therefore, it is crucial to discriminate between different types of predictions, as expectations of different sources may rely on different neural mechanisms and therefore have distinct properties. Similarly, for simple stimuli, such as oriented gratings (*Kok et al., 2012a*; *Kok et al., 2012b*) or simple sequences (*Ekman et al., 2017*), the resolution of expectations may depend less on recurrent processing throughout the visual hierarchy than for complex objects. Thus, it is conceivable that the automaticity of predictive processing partially depends on the complexity of the predictable stimuli and their association, with increasing complexity requiring increasing processing across the hierarchy, and in turn a focus of attention on the predictable stimuli.

## Specific vs. unspecific surprise responses

In LOC and TOFC expectation suppression was largest in neural populations that were driven by the stimuli. Surprisingly, this was not the case in V1, where the suppression was uniformly present in the population that was driven by the stimuli and the population that was not. This replicates the results of our previous study (*Richter et al., 2018*) and suggests that the expectation suppression we observe in V1 is not the result of a stimulus-specific reduction in prediction error responses of neurons processing the stimulus. Rather, they suggest that the observed expectation suppression effect in V1 may be accounted for by a more general response modulation. Widespread nonperceptual modulations of visual cortical activity have been documented in response to unexpected events (*Jack et al., 2006*; *Donner et al., 2008*) and have been suggested to be linked to the cholinergic or noradrenergic system (*Aston-Jones and Cohen, 2005*; *Yu and Dayan, 2005a*). Interestingly, both

the cholinergic and noradrenergic systems have also been associated with fluctuations in pupil dilation (*Reimer et al., 2016*). In line with this, we found a significantly enhanced pupil dilation in response to unexpected stimuli when the objects were attended. This suggests two possible global mechanisms which may partially account for the observed unspecific expectation suppression effect. Given that both pupil dilation (*Reimer et al., 2014*; *Vinck et al., 2015*) and the noradrenergic system (*Berridge et al., 2012*) are associated with arousal changes, it is possible that expectation suppression is partially accounted for by an increased arousal in response to surprising stimuli. A related explanation is that enhanced pupil dilation to surprising stimuli (*Damsma and van Rijn, 2017*; *Kloosterman et al., 2015*; *Preuschoff et al., 2011*) results in enhanced retinal illumination, which in turn leads to stronger responses in early visual areas (*Haynes et al., 2004*), which could potentially also contribute to stimulus unspecific expectation suppression in V1. These interpretations are further supported by the fact that expectation suppression and pupil dilation differences between unexpected and expected attended stimuli were associated, with trailing images that elicit larger pupil dilation differences also showing more pronounced expectation suppression in V1.

It is unlikely however that these explanations can fully account for the observed expectation suppression effect across the visual hierarchy, given the stimulus-specificity of suppression in LOC and TOFC. Also, it is important to bear in mind that earlier studies, using different stimuli and paradigms, did observe stimulus-specific expectation effects in V1 (*Kok et al., 2012a*; *Gavornik and Bear, 2014*). Combined, the evidence suggests that the resolution of prediction errors crucially depends on the visual areas that are specifically coding the feature that is diagnostic of an expectation confirmation or violation, while areas below this level may only witness an unspecific, global modulation in their response, signifying the binary expectation confirmation or violation.

## Attention and prediction errors

Within the predictive coding framework, it has been suggested that attention modulates the gain of prediction error units (*Feldman and Friston, 2010*). On first glance, our results may not appear compatible with the suggestion that attention modulates the gain of prediction errors, because we observe a stimulus-specific bottom-up signal (prediction error) when stimuli are unattended, but no difference in the size of this prediction error between expected and unexpected stimuli. However, it is conceivable that the gain modulation of activity in prediction error units only occurs after the initial feedforward activity sweep, once the object predictions are strongly activated and start exerting an effect on the resolution of the prediction error. In particular, the response to unexpected attended stimuli may be upregulated by attention, while prediction errors for expected attended stimuli are rapidly resolved, thus resulting in the difference in activity for attended objects. On the other hand, when attention is drawn away from the object stimuli, a reduced gain on prediction error units results in the observed attenuation of overall BOLD responses, and an absence of a reliable difference between expected and unexpected stimuli. A closely related, but conceptually distinct, interpretation is that attention constitutes a (modulation of the) prior itself (*Rao, 2005*; *Yu and Dayan, 2005b*). On this account, attention boosts relevant predictions, as during the object classification task, thus leading to wide-spread expectation suppression, due to larger prediction errors for unexpected compared to expected stimuli. However, when attention is disengaged from the object stimuli, object predictions are not generated, and thus do not exert an effect on sensory processing.

## Interpretational limitations

One may wonder whether the character categorization task at fixation may have drawn attention away from the objects so forcefully that the object stimuli were no longer processed by sensory cortex. It is important to note here that, although attention was engaged at fixation by the character categorization task, this task was of trivial difficulty. Thus, it seems unlikely that attentional resources were exhaustively engaged by the task, preventing any processing of the surrounding object stimuli, thereby causing the absence of predictive processing. Indeed, behavioral performance was at ceiling during both tasks. Furthermore, even when objects were unattended reliable visual processing took place, as evident by strong responses and object-specific neural patterns in the visual ventral stream. This suggests that in-depth visual processing of object stimuli did occur in the absence of attention, but predictive processes in particular ceased.

Another alternative explanation of the present results could be that predictive relationships were not learned for the set of objects that were used during the character categorization task, thereby accounting for the absence of a prediction effect. The pair recognition task at the end of the experiment however showed that associations were learned for both image pair sets. Thus, a lack of visual processing or absence of learning cannot account for the observed results. Also, it is worth noting that initially the used probabilistic associations (P(expected|cue)=0.5) may appear less strong than in some previous studies; for example, *Egner et al. (2010)*, *Kok et al. (2012a)*, and *Summerfield et al. (2008)* used P(expected|cue)=0.75. However, the likelihood ratio of expected/unexpected stimuli (0.5/0.1 = 5) used here is actually larger (i.e., each unexpected image is more surprising) than in the cited studies (0.75/0.25 = 3). Moreover, similar probabilistic associations have been successfully employed in studies investigating neural effects of statistical learning in both non-human primates (*Meyer and Olson, 2011*) and humans (*Richter et al., 2018*). In short, the utilized conditional probabilities are comparable to previous studies investigating statistical learning. Finally, it is worth emphasizing that neither adaptation nor familiarity effects can account for the observed results, because all trailing objects served both as expected and unexpected images, depending only on temporal context (i.e., the leading image).

## Conclusion

In sum, our results suggest that visual statistical learning results in attenuated sensory processing for predicted input, but only when this input is attentively processed. Thus, attention seems to gate the integration of prior knowledge and sensory input. This places important constraints on neurocomputational theories that cast perceptual inference as a process of automatic integration of prior and sensory information.

## Materials and methods

### Preregistration and data availability

The present study was preregistered at Open Science Framework (OSF) before any data were acquired. The preregistration document is available at DOI: 10.17605/OSF.IO/36TE7. All procedures and criteria outlined in the preregistration document were followed, unless explicitly specified in the Materials and method section below. In this manuscript, only research question 1 of the preregistration document is addressed. All data analyzed in the present paper are available here: http://hdl.handle.net/11633/aacg3rkw.

### Participants and data exclusion

Our target sample size was n = 34. This sample size was chosen to ensure 80% power for detecting at least a medium effect size (Cohen's d $\geq$ 0.5) with a two-sided paired t-test at an alpha level of 0.05. In total, 38 healthy, right-handed participants were recruited from the Radboud University research participation system. The study followed institutional guidelines of the local ethics committee (CMO region Arnhem-Nijmegen, The Netherlands). We excluded four participants, following our exclusion criteria (see preregistration document and *Data Exclusion*) resulting in the desired sample size of n = 34 participants (25 females, age 24.9 ± 4.8 years, mean ± SD) for data analysis. Of these four exclusions, three exhibited excessive motion during scanning, and one was caused by the participant falling asleep, thus resulting in an incomplete data set.

### Data exclusion

The following preregistered criteria were utilized for the rejection of data. If any of the following criteria applied, data from that participant were excluded from all analyses. (1) Subpar fixation behavior during scanning, indicative by a total duration of closed eyes exceeding 3 SD above the group mean – only trials with stimuli were considered in this analysis; that is, null events and instruction or performance screens were not included. (2) Excessive relative motion larger than ½ voxel size (i.e., 1 mm) during MRI scanning, as indexed by the total number of these motion events exceeding 2 SD above the group mean. (3) Task performance during MRI scanning indicating frequent attentional lapses, as indicated by a mean error rate 3 SD above the group mean.

A fourth rejection criterion, outlined in the preregistration document, based on chance level performance during the post-scan pair recognition task (see: *Pair recognition task* and *2AFC task* in the preregistration document), was not enforced. This decision was based on feedback by participants, indicating that the short ITI during this task made it very challenging, even for participants who reported to have learned most of the associations. Thus, the preregistered pair recognition task based exclusion criterion would not fulfill the desired function of reliably indicating which participants did not explicitly learn the associations, as participants struggled with the task due to its fast pace. Indeed, the enforcement of the criterion would have resulted in the rejection of an additional nine participants (~26% of participants) from data analysis, which was deemed too stringent.

## Stimuli and experimental paradigm

### Experimental paradigm

The experiment consisted of two sessions on two consecutive days. On each day the same stimuli were used for each participant, but different tasks were employed.

*Learning session – day one.* On each trial participants were exposed to two images of objects in quick succession (see *Figure 6A* for a single trial). Each stimulus was presented for 500 ms without an interstimulus interval and an intertrial interval between 1000–2000 ms. Each participant saw 24 different object images, 12 of which only occurred as leading images (i.e., as the first image on a trial), while the remaining 12 occurred only as trailing images (i.e., as the second image on a trial). Importantly, during the learning session the leading image was perfectly predictive of the identity of the trailing image [P(trailing|leading)=1]. In other words, there were 12 image pairs during learning. While participants were made aware of the existence of such regularities, the regularities were not task-relevant. On 20% of trials, one of the two object images was presented upside-down – either the leading or the trailing image could be flipped upside-down. Crucially, whether an image was upside-down could not be predicted and was completely randomized. Participants were instructed to press a button as soon as an upside-down image occurred. Both speed and accuracy were emphasized. On trials without an upside-down image, no response was required. Throughout the entire trial, a fixation bull's-eye (outer circle 0.7° visual angle) was superimposed at the center of the screen. Within the inner circle of the fixation bull's-eye (0.6° visual angle) alphanumeric characters (letters or symbols) were presented (~0.4° visual angle). The characters were presented at the same time and for the same duration as the object stimuli – that is, two characters per trial, each for 500 ms. As with the object images, there were 12 leading characters and 12 trailing characters. However, unlike the objects, the identity of the characters, including whether a letter or symbol occurred, was randomized and thus unpredictable. Participants were instructed that they could ignore these characters, but to maintain fixation on the fixation bulls-eye. In total each participant performed 960 trials during the learning session split into four runs, with a brief break in between runs. Thus, each of the image pairs occurred 80 times during the learning session. The learning session took approximately 60 min. *fMRI session – day two.* Day two of the experiment took place one day after the learning session. First, participants performed an additional 240 trials of the same upside-down task as during the learning session in order to refresh the learned associations. Then participants performed two new tasks in the MRI scanner. During MRI scanning, trials were similar to the learning session, using the same stimulus presentation durations, except for longer intertrial intervals (4000–6000 ms, randomly sampled from a uniform distribution). Another change to the paradigm during MRI scanning was a reduction of the probability of the trailing image given the leading image; P(trailing_expected|leading)=0.5. Thus, now only in 50% of trials a leading image was followed by its expected trailing image. In the remaining 50% of trials, one of the other five trailing images would occur, making these images unexpected given that particular leading image (i.e., each unexpected trailing image had P(trailing_unexpected|leading)=0.1). This was achieved by splitting the original 12 × 12 transition matrix from day one into two 6 × 6 matrices (see *Figure 6B*). One 6 × 6 matrix was used for each of the two tasks participants performed in the MRI (object categorization and character categorization tasks; see below). Thus, each expected trailing image was five times more likely given its leading image than any of the unexpected trailing images. Furthermore, each trailing image was only (un-)expected by virtue of the leading image it followed, which in turn also ensured that all images occurred equally often throughout the experiment, excluding confounds due to stimulus frequency or familiarity. During MRI scanning, an infrared eye tracker (SensoMotoric Instruments, Berlin,

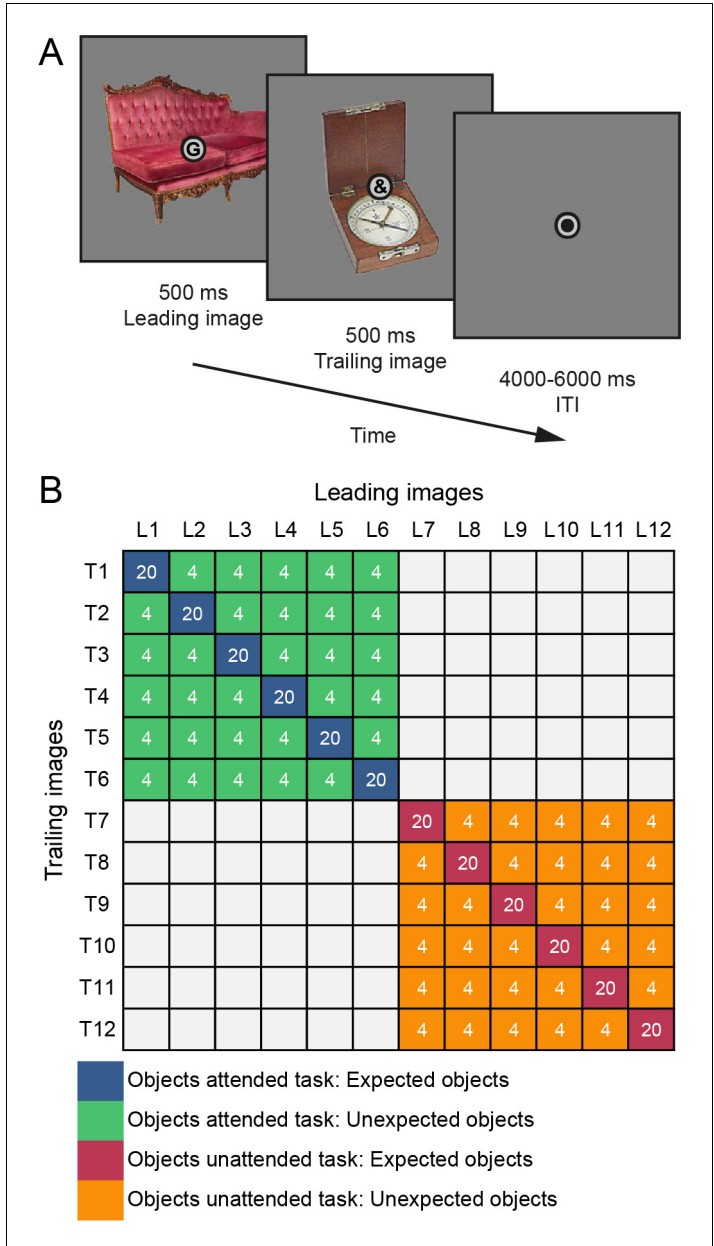

**Figure 6.** Experimental paradigm. (**A**) A single trial is displayed, starting with a 500 ms presentation of the leading object and the leading character superimposed at fixation. Next, without ISI, the trailing object and trailing character are shown for 500 ms. Each trial ends with a 4000–6000 ms ITI (MRI session; 1000–2000 ms ITI learning session), showing only a fixation dot. (**B**) Statistical regularities depicted as image transition matrix with object pairs and trial numbers during MRI scanning. L1 to L12 represent leading objects, while T1 to T12 represent the trailing objects. Leading and trailing objects were randomly selected per participant from a larger pool of images - that is, leading images of one participant may occur as trailing objects of another participant, in a different task, or not at all. Blue cells denote expected object pairs of the objects attended (object categorization) task, while green indicates unexpected object pairs of the objects attended task. Red denotes expected objects of the objects unattended (character categorization) task, and orange indicates unexpected objects of the objects unattended task. Each participant was also assigned 12 leading and 12 trailing characters (six letters, six symbols each). Unlike the object images, there was no association between leading and trailing characters – that is, the identity of the leading and trailing character was unpredictable. White numbers represent the total number of trials of that cell during MRI scanning. In total 120 trials of each of the four conditions were shown during MRI scanning per participant. In the behavioral learning session, participants performed an orthogonal oddball detection task,

*Figure 6 continued on next page*

Figure 6 continued

during which only expected pairs were shown (i.e., only the diagonal of the matrix), for a total of 80 trials per expected pair (960 trials total).

DOI: https://doi.org/10.7554/eLife.47869.015

Germany) was used to monitor and record the position and pupil size of the left eye, at 50 Hz. Finally, after MRI scanning, a brief pair recognition task was performed – for details see *Pair recognition task* below.

*Object categorization task.* During the object categorization task participants were required to categorize, as quickly and accurately as possible, the trailing object on each trial as either electronic or non-electronic. Thus, during this task it was beneficial to be able to predict the identity of the trailing object using the learned associations. Failing to respond, or responding later than 1500 ms after trailing image onset, was considered a miss. Because the 12 × 12 transition matrix was split into two 6 × 6 matrices, one for this task, one for the character categorization task, it was ensured that both 6 × 6 matrices contained three electronic and three non-electronic objects as trailing and leading images, ensuring an equal base rate of both categories. Before performing this task, it was explained that 'electronic' would be any object that contains any electronic components or requires electricity to be used. Furthermore, it was ensured that participants could correctly classify each object by displaying all objects on screen and requesting participants to verbally categorize and name each object before entering the MRI.

*Character categorization task.* Trials of the character categorization task were identical to the object categorization task, except that participants were instructed to categorize the trailing character on each trial as a letter (of the standard Latin alphabet: A, B, D, E, G, H, J, K, M, N, R, S) or non-letter (i.e., a symbol or letter of a non-Latin alphabet: €, $, =, +, Φ, Ɔ, £, ‡, Ϋ, ҍ, !, ?). While the presentation onset and duration of the characters coincided with the presentation of the object images, the identity of the trailing character was not predictable. As with the object images, six characters (three letters, three non-letters) were assigned as leading characters and six were assigned as trailing characters (three letters, three non-letters) for each of the two tasks (object and character categorization task). This was done to ensure that the character categorization task was as similar as possible to the object categorization task, and that exposure to the individual characters was as frequent as to the objects. Thus, in short, the rationale of the character categorization task was to draw attention away from the object stimuli and towards the characters, without imposing a heavy load on attentional or cognitive resources. Indeed, both tasks were designed to yield task performance at ceiling level. For both the object and character categorization tasks, feedback on behavioral performance was provided at the end of each run.

*Procedure, MRI session.* First, participants performed a brief practice run consisting of 50 trials (~5 min in duration) of either the object or character categorization task in the MRI. However, during the practice run, no unexpected trailing images occurred in order to retain the strong expectations built up during the learning session. Additionally, during the practice run, an anatomical image was acquired. After the practice run, two runs of the object or character categorization task were performed. Each run (~14 min) consisted of 120 trials and seven null events of 12 s. Next, a practice run of the other task followed – that is, if the object categorization task was performed first, the character categorization task would now follow, or vice versa. The task order was counter-balanced across participants. The practice run was again followed by two runs of the second task. After this, participants performed one functional localizer run (see: *localizer*). Finally, participants did a pair recognition task (see: *Pair recognition task*), assessing the learning of the object pairs. Once finished, participants were fully debriefed, and any remaining questions were addressed.

*Localizer.* We included a localizer session to define object-selective LOC for each participant and to constrain region of interest (ROI) masks to the most informative voxels using data from an independent, context-neutral run (i.e., without expectations). The functional localizer consisted of a repeated presentation of the previously seen trailing images and their phase-scrambled version. Images were presented for 12 s at a time, flashing at 2 Hz (300 ms on, 200 ms off). At some point during stimulus presentation, the middle circle of the fixation dot would dim. Participants were

instructed to press a button, as fast as possible, once they detected the dimming of the fixation dot. Each trailing image was presented six times. Additionally, a phase-scrambled version of each trailing image was presented three times. Furthermore, 12 null events, each with a duration of 12 s were presented. The presentation order was fully randomized, except for excluding direct repetitions of the same image and ensuring that each trailing image once preceded and once followed a null event in order to optimize the design.

*Pair recognition task.* The rationale of this task was to assess the learning of the object pairs (i.e, statistical regularities) and to compare whether participants learned the regularities during the objects attended task better than during the character categorization task. The pair recognition task followed the MRI session and consisted of the presentation of a leading image followed by two trailing images, one on the left and one on the right of the fixation dot. Participants were instructed to indicate, by button press, which of the two trailing images was more likely given the leading image. In order to prevent extensive learning during this task, a few trials with only unexpected trailing images were shown. Furthermore, participants were instructed that a response was required on each trial, even when they were unsure. Stimulus durations and intertrial intervals were identical to the learning session, that is, 500 ms leading image, 500 ms trailing images, and a variable intertrial interval (1000–2000 ms randomly sampled from a uniform distribution). A response had to be provided within 1500 ms after trailing image onset, or otherwise the trial was counted as a miss. Participants performed one block of this task, consisting of 240 trials.

## Stimuli

Sixty-four full color object stimuli were used during the experiment. The object images were a selection of stimuli from *Brady et al. (2008)* comprising typical object stimuli which were clearly electronic or non-electronic in nature (stimuli can be found here, DOI: 10.17605/OSF.IO/36TE7). Of these 64 object stimuli, 24 were randomly selected per participant, of which 12 were randomly assigned as leading images, while the other 12 served as trailing images. Thus, each specific image could occur as leading image for one participant, as trailing image for another participant, and not at all for a third participant, thereby minimizing the impact of any particular image's features. Images spanned approximately 5° x 5° visual angle on a mid-gray background, both during the learning session and MRI scanning. During the learning session stimuli were presented on an LCD screen (BenQ XL2420T, 1920 × 1080 pixel resolution, 60 Hz refresh rate). During MRI scanning, stimuli were back-projected (EIKI LC-XL100 projector, 1024 × 768 pixel resolution, 60 Hz refresh rate) on an MRI-compatible screen, visible using an adjustable mirror mounted on the head coil.

We calculated the average relative luminance of the object stimuli by converting the stimulus images from sRGB to linear RGB, then calculated the relative luminance for all pixels (where relative luminance $Y = 0.2126*R + 0.7152*G + 0.0722*B$; *Stokes et al., 1996*), and finally averaged the obtained luminance values, thereby obtaining the mean relative luminance per image. On this relative luminance scale, 0 would be a completely black image, while one would be a white image. The average relative luminance of the stimulus set was 0.225, while the relative luminance of the mid gray background, presented during the ITI, was 0.216.

## fMRI data acquisition

Anatomical and functional images were acquired on a 3T Prisma scanner (Siemens, Erlangen, Germany), using a 32-channel head coil. Anatomical images were acquired using a T1-weighted magnetization prepared rapid gradient echo sequence (MP-RAGE; GRAPPA acceleration factor = 2, TR/TE = 2300/3.03 ms, voxel size 1 mm isotropic, 8° flip angle). Functional images were acquired using a whole-brain T2*-weighted multiband-6 sequence (time repetition [TR]/time echo [TE]=1000/34.0 ms, 66 slices, voxel size 2 mm isotropic, 75° flip angle, A/P phase encoding direction, FOV = 210 mm, BW = 2090 Hz/Px). To allow for signal stabilization, the first five volumes of each run were discarded.

## Data analysis
### Behavioral data analysis
Behavioral data from the main task MRI runs were analyzed in terms of reaction time (RT) and response accuracy. Trials with RT <200 ms, RT >1500 ms, or no response were rejected as outliers

from RT analysis (1.56% of trials). The two factors of interest were expectation status (expected vs. unexpected) and attention (objects attended vs. objects unattended task). Thus, a 2 × 2 repeated measures analysis of variance (RM ANOVA) was used to analyze behavioral data, with the additional planned simple main effects analyses of expected vs. unexpected within each task condition using two-sided paired t-tests. For these tests, RT and accuracy data per participant were averaged across trials and subjected to the analyses. For all paired t-tests, the effect size was calculated in terms of Cohen's $d_z$ (*Lakens, 2013*), while partial eta-squared ($\eta^2$), as implemented in JASP (*JASP Team, 2018*), was used as a measure of effect size for the RM ANOVA. Standard errors of the mean were calculated as the within-subject normalized standard error of the mean (*Cousineau, 2005*) with bias correction (*Morey, 2008*). Data from the pair recognition task were analyzed by means of two-sided paired t-tests or Wilcoxon signed rank test, if the normality assumption was violated, comparing RTs and response accuracies between image pairs belonging to the attended vs. unattended conditions. Effect size for Wilcoxon signed rank test was calculated as the matched rank biserial correlation ($r_b$; *JASP Team, 2018*).

## fMRI data preprocessing

fMRI data were preprocessed using FSL 5.0.11 (FMRIB Software Library; Oxford, UK; www.fmrib.ox.ac.uk/fsl; *Smith et al., 2004*, RRID:SCR_002823). The preprocessing pipeline consisted of the following steps: brain extraction (BET), motion correction (MCFLIRT), grand mean scaling, temporal high-pass filtering (128 s). For univariate analyses, data were spatially smoothed (Gaussian kernel with full-width at half-maximum of 5 mm), while for multivariate analyses no spatial smoothing was applied. FSL FLIRT was used to register functional images to the anatomical image (BBR) and the anatomical image to the MNI152 T1 2 mm template brain using linear registration (12 degrees of freedom). Registration to the MNI152 standard brain was only applied for whole-brain analyses, while all ROI analyses were performed in each participant's native space in order to minimize data interpolation.

## fMRI data analysis

FSL FEAT was used to fit voxel-wise general linear models (GLM) to each participant's run data in an event-related approach. In these first-level GLMs, expected and unexpected image pair events were modeled as two separate regressors with a duration of one second (the combined duration of leading and trailing image) and convolved with a double gamma haemodynamic response function. An additional regressor of no interest was added to the GLM, modeling the instruction and performance summary screens. Moreover, the first temporal derivatives of these three regressors were added to the GLM. Finally, 24 motion regressors (FSL's standard + extended set of motion parameters) were added to account for head motion, comprised of the six standard motion parameters, the squares of the six motion parameters, the derivatives of the standard motion parameters and the squares of the derivatives. The contrast of interest, expectation suppression, was defined as the BOLD response to unexpected minus expected images. FSL's fixed effects analysis was used to combine data across runs. Because each run either used the objects attended or objects unattended (character categorization) task, two separate regressors were used in the fixed effects analysis, one for the objects attended task, one for the objects unattended task. Finally, across participants, data were combined using FSL's mixed effects analysis (FLAME 1). Gaussian random-field cluster thresholding was used to correct for multiple comparisons, using the updated default settings of FSL 5.0.11, with a cluster formation threshold of p<0.001 (one-sided; that is, $z \geq 3.1$) and cluster significance threshold of p<0.05.

## Region of interest (ROI) analysis

ROI analyses were conducted in each participant's native space. The three a priori defined and pre-registered ROIs were V1, object-selective LOC and TOFC. The choice of these ROIs was based on our previous study (*Richter et al., 2018*), in which we found significant expectation suppression in these cortical areas. For each ROI the mean parameter estimate was extracted from the participant's parameter estimate maps, representing the expected and unexpected images. This was done separately for the objects attended and objects unattended tasks, thus resulting in four parameter of interest. The parameter estimates were divided by 100 to yield percent signal change relative to baseline (*Mumford, 2007*). For each ROI, these data were submitted to a 2 × 2 RM ANOVA with

expectation (expected, unexpected) and attention (objects attended, objects unattended) as factors. Simple main effects were calculated for the expectation effect in each of the attention conditions using two-sided paired t-tests. As applicable, Cohen's $d_z$ or partial eta-squared ($\eta^2$) were calculated as measures of effect size. Again, the within-subject normalized standard error of the mean (*Cousineau, 2005*) with bias correction (*Morey, 2008*) was calculated as an indicator of the standard error.

*ROI definition.* All ROIs were preregistered and defined a priori, based on previous results, and refined using independent data. The three ROIs were V1, object selective LOC, and TOFC. V1 was defined based on each participant's anatomical image, using Freesurfer 6.0 for cortex segmentation (recon-all; *Dale et al., 1999*; RRID:SCR_001847). The resulting V1 labels were transformed into native volume space using 'mri_label2vol' and merged into one bilateral mask. LOC masks were created in each participant's native space using data from the functional localizer. Object selective LOC was defined as bilateral clusters, within anatomical LOC, showing a significant preference for intact compared to scrambled object stimuli (*Kourtzi and Kanwisher, 2001*; *Haushofer et al., 2008*). To this end, one regressor modeling intact objects and one regressor modeling scrambled objects were fit to each participant's localizer data. Additional regressors of no interest were added to the model, with one regressor modeling instruction and performance screens, the temporal derivatives of all regressors, and the 24 motion regressor as also described above (see: *fMRI data analysis*). The contrast of interest, objects minus scrambles, was constrained to anatomical LOC, and the largest contiguous clusters in each hemisphere were extracted per participant. By default, the contrast was thresholded at $z >= 5$ (uncorrected; that is, $p<1e-6$). The threshold was lowered on a per participant basis if the resulting LOC clusters were too small; that is, bilateral mask with less than 400 voxels in native volume space. The TOFC ROI mask was created using an anatomical temporal-occipital fusiform cortex mask from the Harvard-Oxford cortical atlas (RRID:SCR_001476), as distributed with FSL. This mask was further constrained to voxels showing a significant expectation suppression effect on the group level in our previous study, using an independent data set (Figure 2A in *Richter et al., 2018*). The resulting mask was transformed from MNI space to each participant's native space using FSL FLIRT.

Finally, each of the three ROI masks were constrained to the 300 voxels forming the most informative neighborhoods concerning object identity decoding. This was done by performing a multi-voxel pattern analysis (see: *Multi-voxel pattern analysis (MVPA)*) on the localizer data set per participant, decoding object identity. This ensured that the final masks contained the voxels that were from the most informative neighborhoods in each respective mask. It was not required that the final mask formed one contiguous cluster. In order to verify that our results did not depend on the a priori defined but arbitrary number of voxels in the ROI masks, we repeated all ROI analyses with masks ranging from 100 to 400 voxels (i.e., 800 mm$^3$ to 3200 mm$^3$) in steps of 100 voxels.

## Multi-voxel pattern analysis (MVPA)

A decoding analysis was performed on each participant's localizer data. For this analysis, not spatially smoothed mean parameter estimate maps were obtained per localizer trial by fitting a GLM with only one trial as regressor of interest and all remaining trials as one regressor of no interest (*Mumford et al., 2012*). Subsequently, these parameter estimate maps were used in a multi-class, linear SVM-based decoding analysis (SVC function, Scikit-learn; *Pedregosa et al., 2011*; RRID:SCR_002577), with the 12 trailing images as classes. The analysis was performed on the localizer data across the whole brain using a searchlight approach (6 mm radius) and stratified 4-fold cross-validation. Finally, the resulting decoding accuracy maps were used to constrain the ROI masks (see *ROI definition*).

We employed a similar decoding analysis to determine whether object-specific neural activity in the visual ventral stream was equally present during both the objects attended and unattended tasks. As above, a multi-class decoder with linear SVMs was used to decode object images. The per trial parameter estimates of the localizer run served as training data. For each main task run voxel-wise GLMs were fit with a regressor for each trailing image per expectation condition. As in the other fMRI analyses, the 24 motion regressors and temporal derivatives were added to the model (see *fMRI data analysis*). Finally, the decoder was tested on the obtained trailing image parameter estimates per run. As each attention condition consisted of six trailing images, chance performance of this decoder was at 16.7% (1/6).

## Stimulus specificity analysis

In an effort to further explore the nature of expectation suppression throughout the ventral visual stream, we investigated the stimulus specificity of the suppression effect. The key question here was if expectation suppression was primarily present in stimulus-driven voxels within a given area, or whether most voxels in an area showed the effect, regardless of whether or not they were stimulus-driven.

In order to investigate specificity, we obtained anatomically defined masks of our three ROIs (V1, LOC, TOFC). For V1 the unconstrained, anatomically defined Freesurfer V1 mask was used (see *ROI definition*). Anatomical LOC and TOFC were defined using the Harvard-Oxford cortical atlas. FSL FAST was used to obtain a gray matter mask for each participant based on their anatomical scan. Masks were transformed to the participant's native EPI space. Next, the three ROI masks were constrained to the participant's gray matter voxels. Within the resulting ROI masks, using the contrast object stimuli compared to baseline from the functional localizer run, voxels were split into two categories, stimulus-driven ($z > 1.96$; that is, $p < 0.05$, two-sided), and not stimulus-driven, but also not deactivated, voxels ($-1.96 < z < 1.96$). Average expectation suppression was compared between ROIs split into stimulus-driven vs. not stimulus-driven voxels. Thus, a $3 \times 2$ RM ANOVA with ROI (V1, LOC, TOFC) and stimulus-driven (stimulus-driven vs not stimulus-driven) as factors was used for analysis. Greenhouse-Geisser correction was applied, if Mauchly's sphericity test indicated a violation of the sphericity assumption. Furthermore, the simple main effect of stimulus-driven vs. not stimulus-driven was assessed within each ROI. Additionally, to test for the presence of any expectation suppression, the amount of suppression was compared against zero using one sample t-tests.

## Pupillometry

In order to investigate whether pupil dilation effects accompany expectation suppression, we analyzed the pupil diameter data recorded during MRI scanning. A priori, two participants were rejected from this analysis, as the experiment log book indicated that pupil diameter data were unreliable for these two participants, leaving 32 participants for pupillometry. First, blinks were detected using a velocity based method, following the procedure outlined by *Mathôt (2013)*. A blink was defined as a negative velocity peak (eyes closing), followed by a positive velocity peak (eyes opening) within a time period of 500 ms. The velocity threshold was set to 5 (arbitrary units). An additional 100 ms were added as padding before and after the detected blink onset and offset. If padding resulted in overlapping blink windows, consecutive blinks were considered as one long blink. Linear interpolation was used to replace missing data during blinks (18.05% of data). Note, this number includes the padding, and all time periods of no interest, such as null events, instruction and performance screens, as well as recording periods before and after MRI run onset; that is, periods during which participants were free to close their eyes. Remaining missing data, not following a typical blink profile, were excluded from analysis, again adding a padding of 100 ms (3.07% of data). Similarly, outlier data with implausible velocity profiles were also rejected from the analysis, using the same velocity-based threshold as for blink detection but without the criterion of a negative peak followed by a positive peak (5.30% of data). Thus, data interpolation was only applied for short time intervals, which represent a clear blink, in order to avoid interpolation based on artifacts or over exceedingly long time periods. Finally, pupil data were smoothed using a Hanning window of 200 ms, and epoched into trials from 1 s before trailing image onset to 4 s after trailing image onset. The data of each trial were baseline corrected by diving the pupil diameter estimates by the mean diameter during the baseline period, 0.5 to 0 s before leading image onset. As a final data quality check, all trials exceeding pupil diameter values 7 SDs above the mean pupil diameter were rejected (3.01% trials). Trials with expected trailing images and unexpected trailing images were averaged separately for each participant. The difference between unexpected minus expected was subjected to a cluster-based permutation test (100,000 permutations; two-sided $p < 0.05$; cluster formation threshold $p < 0.05$) in order to assess statistical significance. Data from the objects attended and the objects unattended tasks were analyzed separately.

## Linking pupil and neural measures

In an exploratory analysis we sought to provide additional evidence for an association between pupil dilation and expectation suppression. To this end, we correlated expectation suppression with pupil

dilation differences between expected and unexpected objects per trailing image. First, we obtained per trailing image parameter estimates by fitting a voxel-wise GLM to the fMRI data for each run, following the same procedure as for the main fMRI data analysis, outlined in *fMRI data analysis* and *Region of interest (ROI) analysis*. The only difference was that a separate regressor per trailing image and expectation condition was fit, thus resulting in a model with 12 regressors of interest (six trailing images * two expectation conditions). As before, data were combined across runs using FSL's fixed effect analysis. The resulting parameter estimate maps were extracted for each ROI (V1, LOC, TOFC) and converted to percent signal change. Finally, for each participant we calculated expectation suppression for each trailing image (expectation suppression = $BOLD_{unexpected} - BOLD_{expected}$). Similarly, we calculated the difference in pupil dilation between unexpected and expected occurrences of each trailing image. For this we extracted the preprocessed (see: *Pupillometry*) pupil size estimates for each trial and calculated the mean pupil size within the time window that showed a significant difference in pupil dilation between unexpected compared to expected attended stimuli on the group level (*Figure 3*, left panel); that is, 1.52 to 2.88 s after trailing image onset. Next, we calculated the average difference in pupil size for each trailing image for unexpected compared to expected occurrences, thus yielding six pupil size difference scores (unexpected − expected) for both attention tasks per participant. Spearman's rank correlation was then used to estimate the correlation between the pupil dilation differences and expectation suppression magnitudes for each participant. Therefore, this correlation expresses the correlation in ranks of pupil dilation differences and expectation suppression magnitude for the trailing images, with positive correlations indicating that trailing images with large expectation suppression effects are also associated with larger pupil dilation differences. The obtained correlation coefficients were Fisher z-transformed and compared against zero (no correlation) using one-sample t-tests for each ROI and attention condition. We also submitted the coefficients to a repeated measures ANOVA with ROI and attention as factors.

## Linking behavioral and neural measures

In another exploratory analysis we investigated the relationship between behavioral and neural benefits of expectations by correlating expectation suppression with the behavioral RT benefit for expected stimuli observed during MRI scanning. First, we calculated the RT benefit for each trailing image during the main fMRI task ($RT_{benefit} = RT_{unexpected} - RT_{expected}$, per trailing image). Within each ROI we then correlated expectation suppression per trailing image (see: *Linking pupil and neural measures*) with RT benefit per trailing image using Spearman's rank correlation. Thus, this correlation coefficient indicates the rank correlation of expectation induced RT benefits and expectation suppression magnitude for the different trailing images. For statistical inference across participants, we Fisher z-transformed the correlation coefficients, and tested whether the observed correlation coefficients differ from zero (no correlation) in each condition by performing one-sample t-tests for each ROI and attention task separately. Finally, we also compared the magnitude of the correlations between ROIs and attention tasks using a $3 \times 2$ repeated measures ANOVA with ROI and attention condition (task) as factors.

## Bayesian analyses

In order to further evaluate any non-significant tests, in particular simple main effects, we performed the Bayesian equivalents of the above outlined analyses. JASP 0.9.0.1 (*JASP Team, 2018*; RRID: SCR_015823) was used to perform all Bayesian analyses, using default settings. Thus, for Bayesian t-tests a Cauchy prior width of 0.707 was chosen. Qualitative interpretations of Bayes Factors are based on criteria by *Lee and Wagenmakers (2013)*.

## Software

MRI data preprocessing and analysis was performed using FSL 5.0.11 (FMRIB Software Library; Oxford, UK; www.fmrib.ox.ac.uk/fsl; *Smith et al., 2004*; RRID:SCR_002823). Custom Python 2.7.13 (Python Software Foundation, RRID:SCR_008394) scripts were used for additional analyses, data handling, statistical tests and data visualization. The following Python libraries and toolboxes were used: NumPy 1.12.1 (*van der Walt et al., 2011*; RRID:SCR_008633), SciPy 0.19.0 (*Jones et al., 2001*; RRID:SCR_008058), Matplotlib 1.5.1 (*Hunter, 2007*; RRID:SCR_008624), Statsmodels 0.8.0 (www.statsmodels.org) and Scikit-learn 0.18.1 (*Pedregosa et al., 2011*; RRID:SCR_002577).

Additionally, Slice Display (*Zandbelt, 2017*), a MATLAB 2017a (The MathWorks, Inc, Natick, Massachusetts, United States, RRID:SCR_001622) data visualization toolbox, was used for displaying whole-brain results. JASP 0.9.0.1 (*JASP Team, 2018*; RRID:SCR_015823) was used for Bayesian analyses and RM ANOVAs. Stimuli were presented using Presentation software (version 18.3, Neurobehavioral Systems, Inc, Berkeley, CA, RRID:SCR_002521).

## Supplemental analyses

### Pupil dilation is associated with larger BOLD responses

In order to provide additional support for the hypothesis that pupil dilation differences may partially underlie expectation suppression in V1, we examined the relationship between pupil dilation and the BOLD response. First, we extracted per trial pupil size data and parameter estimate maps from the fMRI main task data for V1. Pupil size data were preprocessed as described in *Pupillometry*, and extracted from a three-second time window, starting with trailing image onset and ending 2.5 s after trailing image offset; thus, the time window covered the full duration shown in *Figure 3* after trailing image onset. fMRI data were preprocessed as outlined in *fMRI data preprocessing*. Next, for each trial we fitted a GLM with only one trial as regressor of interest and all remaining trials as regressors of no interest (*Mumford et al., 2012*). Per participant, we extracted the per trial parameter estimate maps, averaged within the V1 ROI, and z scored the mean parameter estimates per condition separately in order to remove potential effects of mean differences between the conditions. We also z scored the pupil size estimates per condition for the same reason. Next, we fitted per participant a GLM with the mean BOLD parameter estimates (one per trial) as predicted variable and a regressor with pupil size for each expectation and attention condition combination (i.e., four regressors of interest) as predictors. Statistical inference across subjects was performed by subjecting the thus obtained parameter estimates of the four regressors of interest to a 2 × 2 repeated measures ANOVA, as with our main ROI analysis; that is, with attention and expectation as factors. Furthermore, in order to assess whether the BOLD response was influenced by pupil dilation at all we performed one-sample t-tests comparing the obtained parameter estimates against zero for each condition separately. Additionally, we performed a similar analysis, but split the fMRI data into stimulus-driven vs. non-stimulus-driven V1 gray matter voxels (see *Stimulus specificity analysis* for details on the ROI mask creation). This analysis thus results in a 2 × 2 × 2 repeated measures ANOVA with expectation, attention and stimulus-responsiveness as factors.

Increased pupil dilations were associated with larger BOLD responses regardless of whether stimuli were attended and expected (attended expected: $t_{(31)}$ = 3.006, p=0.005, $d_z$ = 0.531; attended unexpected: $t_{(31)}$ = 4.392, p=1.2e-4, $d_z$ = 0.776; unattended expected: $t_{(31)}$ = 5.228, p=1.1e-5, $d_z$ = 0.924; unattended unexpected: $W$ = 452, p=2.1e-4, $r_B$ = 0.712). Results are shown in *Figure 3—figure supplement 1*. Pupil dilation led to slightly stronger BOLD increases when objects were unattended than attended ($F_{(1,31)}$ = 5.563, p=0.025, $\eta^2$=0.152), but independent of whether stimuli were expected or unexpected ($F_{(1,31)}$ = 0.054, p=0.817, $\eta^2$=0.002; interaction: $F_{(1,31)}$ = 2.261, p=0.143, $\eta^2$=0.068). Thus, pupil dilation had a positive effect on overall BOLD responses in V1.

### Pupil dilation influences BOLD responses more in non-stimulus-driven V1 voxels

Next, we assessed whether the same association would hold in stimulus-driven and non-stimulus driven V1 voxels. *Figure 3—figure supplement 2* shows that there was indeed a reliable, positive association between BOLD responses and pupil dilation within both stimulus-driven and non-stimulus-driven voxels. Again, larger pupil dilations were predictive of enhanced BOLD responses when object stimuli were attended and unattended, as well as for expected and unexpected objects (all t-tests p<0.05). Interestingly, this association was somewhat larger in non-stimulus-driven than stimulus-driven voxels ($F_{(1,31)}$ = 9.267, p=0.005, $\eta^2$=0.230), suggesting that the association between BOLD and pupil dilation is particularly strong for those neural populations that are not driven by our object stimuli. This is in line with earlier observations that non-stimulus-driven activations (possibly reflecting neuromodulation) are greater in regions that represent more peripheral parts of the visual field (*Jack et al., 2006*). There was also a stronger association of pupil dilation and BOLD responses when objects were unattended ($F_{(1,31)}$ = 5.042, p=0.032, $\eta^2$=0.140), but the magnitude of the

association was not affected by whether a stimulus was expected or not ($F_{(1,31)}$ = 0.008, p=0.928, $\eta^2$=2.6e-4). Moreover, no interaction effect was observed (all interactions p>0.1).

Thus, to summarize, our results show that pupil dilation has a substantial, positive association with V1 BOLD responses, regardless of whether stimuli were attended and expected, for both stimulus-driven and non-stimulus-driven neural populations. This result is expected, given that pupil dilation has been related to other processes known to correlate with BOLD responses such as mental effort, arousal and attention (for a review see: *Mathôt, 2018*). Moreover, increases in retinal illumination due to larger pupil dilation can also result in increased BOLD activity (*Haynes et al., 2004*). These results support our suggestion that larger pupil dilations in response to unexpected stimuli, possibly reflecting general arousal mechanisms, may partially account for expectation suppression in V1. However, it should also be noted that the association between pupil dilation and the BOLD response is not solely observed when objects were attended, as pupil dilation is likely a general reflection of vigilance and arousal (*Reimer et al., 2014*; *Vinck et al., 2015*), which is expected to fluctuate also when the objects are not attended. That this association is more pronounced in non-stimulus-driven voxels, further supports the possibility that expectation suppression in V1, including the suppression observed in non-stimulus-driven voxels, may partially reflect non-perceptual effects such as arousal changes, which are reflected by larger pupil dilations in response to surprising stimuli.

## No differences in pupil dilation during baseline

We assessed pupil size during baseline to ensure that differences in pupil dilation between expectation conditions or attention tasks do not simply reflect difference in baseline (e.g., pre-stimulus arousal differences). Pupil data were preprocessed using the same pipeline as described in *Pupillometry*, except for that pupil size was extracted in raw units during the baseline period. Per participant, pupil size was then averaged for each attention and expectation condition separately. Mean pupil estimates were then compared between conditions using a 2 × 2 repeated measures ANOVA, with expectation and attention as factors. Additionally, a Bayesian repeated measure ANOVA was conducted to quantify the evidence for the absence of a difference in pupil size during baseline.

Results showed that there was no difference in baseline pupil size before attended compared to unattended stimuli ($F_{(1,31)}$ = 5.226, p=0.484, $\eta^2$=0.016, $BF_{inclusion}$ = 0.254), nor before expected compared to unexpected stimuli ($F_{(1,31)}$ = 0.001, p=0.926, $\eta^2$=2.8e-4, $BF_{inclusion}$ = 0.136; interaction: $F_{(1,31)}$ = 6.2e-4, p=0.955, $\eta^2$=1.0e-4, $BF_{inclusion}$ = 0.042). *Figure 3—figure supplement 3* shows the pupil size in raw units during the baseline period. Thus, data suggest that pupil size, and thereby likely arousal, during baseline was of a similar magnitude during both attention tasks and expectation conditions, thereby rendering an explanation of the observed phasic differences in pupil size based on differences in baseline pupil size unlikely.

## Acknowledgements

We thank Matthias Ekman, Mariya Manahova, Peter Kok and Karl Friston for helpful comments and discussions of the manuscript and results. We thank José Marques for assistance with MR sequence questions.

## Additional information

### Competing interests

Floris P de Lange: Reviewing editor, *eLife*. The other author declares that no competing interests exist.

### Funding

| Funder | Grant reference number | Author |
| --- | --- | --- |
| Nederlandse Organisatie voor Wetenschappelijk Onderzoek | Vidi Grant 452-13-016 | Floris P de Lange |

| | | |
|---|---|---|
| Horizon 2020 Framework Programme | ERC Starting Grant 678286 | Floris P de Lange |

The funders had no role in study design, data collection and interpretation, or the decision to submit the work for publication.

### Author contributions
David Richter, Conceptualization, Resources, Data curation, Software, Formal analysis, Validation, Investigation, Visualization, Methodology, Writing—original draft, Project administration, Writing—review and editing; Floris P de Lange, Conceptualization, Formal analysis, Supervision, Funding acquisition, Writing—review and editing

### Author ORCIDs
David Richter https://orcid.org/0000-0002-3404-8374
Floris P de Lange http://orcid.org/0000-0002-6730-1452

### Ethics
Human subjects: The study followed institutional guidelines of the local ethics committee (CMO region Arnhem-Nijmegen, The Netherlands; Protocol CMO2014/288), including informed consent of all participants.

### Decision letter and Author response
Decision letter https://doi.org/10.7554/eLife.47869.021
Author response https://doi.org/10.7554/eLife.47869.022

## Additional files

### Supplementary files
• Supplementary file 1. Overview of expectation suppression across cortex. Brain areas showing significant expectation suppression (GRF cluster corrected). Listed are significant clusters with their respective area label, MNI coordinate of the peak z value, the number of voxels in the cluster, as well as the p value of the cluster and its max z statistic. For large clusters (n voxels >700) additional local z maxima ($z > 3.72$; that is, p<0.0001, one-sided) are also shown with area label, MNI coordinates and max z statistic. Unexp. = unexpected image pairs; Exp. = expected image pairs; Att. = objects attended task; Unatt. = objects unattended (characters attended) task.
DOI: https://doi.org/10.7554/eLife.47869.016

• Transparent reporting form
DOI: https://doi.org/10.7554/eLife.47869.017

### Data availability
All data and code necessary to replicate the reported results are available via the following URL: http://hdl.handle.net/11633/aacg3rkw.

The following dataset was generated:

| Author(s) | Year | Dataset title | Dataset URL | Database and Identifier |
|---|---|---|---|---|
| Richter D, de Lange FP | 2019 | Attentional modulation of perceptual predictions | http://hdl.handle.net/11633/aacg3rkw | Donders Institute, 11633/aacg3rkw |

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
