## [Decision Letter]

Thank you for submitting your article "Statistical learning attenuates visual activity only for attended stimuli" for consideration by *eLife*. Your article has been reviewed by three peer reviewers, and the evaluation has been overseen by a Reviewing Editor and Michael Frank as the Senior Editor. The following individuals involved in review of your submission have agreed to reveal their identity: Valentin Wyart (Reviewer #1); Jakob Heinzle (Reviewer #3).

The reviewers have discussed the reviews with one another and the Reviewing Editor has drafted this decision to help you prepare a revised submission.

Summary:

The study described in this manuscript provides evidence that statistical learning does not (automatically) lead to suppression of expected stimuli when stimuli are not attended (i.e. not task relevant). This finding is supported by a series of clever experiments (fMRI, behavior, pupillometry) and analysis (classical GLM based brain mapping, ROI based analysis, MVPA, behavioral tests) that rule out several potential confounds. The key result is that sensory attenuation disappears when attention is diverted to a second task at fixation. The authors interpret their findings in light of existing theories of predictive processing.

All reviewers agreed that your paper addresses a novel, interesting and important question, and that the results provide important evidence informing models of predictive coding. Reviewers also raised a few points that should be addressed in a revision. These include additional analyses liking pupil and fMRI data, additional information about the stimulus material and pupil response, and the interpretation of the results.

Essential revisions:

1) The authors suggest, based on effects found in pupil dilation (larger phasic dilation for surprising object stimuli when attended), that some of the neural effects found (in particular in V1) may be caused by an increased arousal for surprising stimuli. This relationship between arousal and the neural effects observed in the fMRI data occupies a central space in the Discussion section, but it is currently not well supported by analyses that directly link the pupil and fMRI data. It would be important to test more directly the relationship between (pupil-linked) arousal and the attenuation of sensory responses for expected stimuli (or the enhancement of sensory responses for surprising stimuli). For instance, the authors could perform a mediation analysis by using the phasic pupil dilation presented in Figure 3 as a modulator of the difference between expected and unexpected trailing object stimuli in the BOLD signals. One would predict that a larger pupil dilation may increase overall sensory responses to both expected and surprising stimuli, or maybe even only to surprising stimuli. Another prediction would be that this relationship between pupil dilation and BOLD effects is reduced (or even absent) when the object stimuli are not attended. The authors further propose that stimulus-specific and stimulus-unspecific attenuations may rely on different mechanisms (the stimulus-unspecific attenuation being driven primarily by arousal). Using the mediation analysis suggested above, the authors could directly test such interpretation. Alternatively, they could bin the trials based on pupil dilation response and test whether the BOLD suppression effects depend on pupil-linked neuromodulation. In addition, it could be informative to know whether the neural (and pupil) effects predict behavioral performance (although this may be difficult because behavioral performance was at ceiling in the attended condition). In any case, additional analyses linking pupil data to behavioral data and/or fMRI would be helpful for supporting the conclusions.

2) Reviewers were surprised that no constriction (pupil light reflex) was found at stimulus onset, which presumably involves increases in contrast and light. Were the stimuli and ITI period matched in luminance? They for sure differ in contrast, to which the pupil also responds. Was the screen was quite dim in the scanner, so no transient constriction was elicited? Or was it perhaps that due to the quite narrow ITI range trial onset was quite predictable? Also, baseline pupil should be tested for the 2 tasks (e.g. the raw units) to see whether baseline arousal is different between conditions. This might help elucidate the phasic differences as well, since a lower baseline might be associated with a higher phasic response. Also, the increased dilation comes very late, when the pupil is returning to baseline, suggesting prolonged processing for unexpected stimuli. The baseline was taken from -1 s, giving the pupil response only 4 s to die out for the shortest ITI, which is quite short. The authors could control for expectedness in the previous trial to take into account the supposedly higher baseline after an unexpected trial.

3) There are several points that need a more thorough discussion.

a) While the authors frame their study as an attention study, one could also consider the finding an effect of task (task relevant vs. task irrelevant). Of course, the two concepts overlap, but this should be discussed.

b) Is the explanation for the discrepancy with other studies related to differently controlled attention (as suggested by the authors) or, possibly, to differences in the nature of the stimuli and differences in how they are processed? Importantly, the statistical learning in this study was not automatic (even if being engaged in a different task participants were informed about the regularities) and thus it is likely that they paid attention to the regularities (the other task was easy – performance at ceiling), which might be a crucial difference to other settings. For example, there are auditory, but also visual, mismatch negativity studies which show mismatch effects when the stimuli are not attended. It seems highly unlikely that all of these studies did not control correctly for attention.

c) The implications of the current findings for models of predictive coding need a more balanced discussion. While reviewers agree that the current evidence is not strongly supportive of the idea that attention provides gain control on prediction errors, their main argument in support of this may not be fully correct: It is well conceivable that there is very strong gain control on prediction error neurons (possibly a small subset of all neurons) even if there is strong baseline activity in response to all stimuli. Why should gain control affect all neurons in a region and not be neuron specific?

---

## [Author Response]

Essential revisions:1) The authors suggest, based on effects found in pupil dilation (larger phasic dilation for surprising object stimuli when attended), that some of the neural effects found (in particular in V1) may be caused by an increased arousal for surprising stimuli. This relationship between arousal and the neural effects observed in the fMRI data occupies a central space in the Discussion section, but it is currently not well supported by analyses that directly link the pupil and fMRI data. It would be important to test more directly the relationship between (pupil-linked) arousal and the attenuation of sensory responses for expected stimuli (or the enhancement of sensory responses for surprising stimuli).For instance, the authors could perform a mediation analysis by using the phasic pupil dilation presented in Figure 3 as a modulator of the difference between expected and unexpected trailing object stimuli in the BOLD signals. One would predict that a larger pupil dilation may increase overall sensory responses to both expected and surprising stimuli, or maybe even only to surprising stimuli. Another prediction would be that this relationship between pupil dilation and BOLD effects is reduced (or even absent) when the object stimuli are not attended. The authors further propose that stimulus-specific and stimulus-unspecific attenuations may rely on different mechanisms (the stimulus-unspecific attenuation being driven primarily by arousal). Using the mediation analysis suggested above, the authors could directly test such interpretation. Alternatively, they could bin the trials based on pupil dilation response and test whether the BOLD suppression effects depend on pupil-linked neuromodulation.

We agree that establishing an association between pupil dilation and BOLD responses is important for our interpretation that pupil dilation differences may underlie expectation suppression in V1. We followed the reviewer’s suggestions and explored the relationship between pupil dilation and BOLD responses more directly. Specifically, we would expect larger BOLD responses to be associated with larger pupil dilations. We indeed find a positive association between pupil dilation and the BOLD response in V1, regardless of whether stimuli were attended and expected (all *p*<0.005). We also considered the reviewer’s suggestion of investigating the association of pupil dilation and BOLD responses within stimulus-driven compared to non-stimulus-driven voxels. One would expect the general relationship between pupil dilation and BOLD to be similar in these neural populations, given that arousal-related mechanisms are assumed to effect neural populations in a retinotopically unspecific manner. We do indeed find a positive association between BOLD and pupil dilation in both neural populations with a larger effect in non-stimulus-driven populations.

We added the following subsections to a new “Supplemental analyses” section in the manuscript (“Pupil dilation is associated with larger BOLD responses” and “Pupil dilation influences BOLD responses more in non-stimulus-driven V1 voxels”

After establishing a reliable association between pupil dilation and BOLD responses, we investigated whether there is evidence for an association between pupil dilation differences between expected compared to unexpected stimuli and expectation suppression (i.e. BOLD_unexpected_ – BOLD_expected_). We now describe this additional analyses in the Materials and methods subsection “Linking pupil and neural measures”.

The results of this analysis, and a new Figure 4A, are now presented in the Results subsection “Expectation suppression and pupil dilations to surprising stimuli are associated”.

Finally, the new results are briefly discussed in the paragraph discussing the pupil dilation differences in the Discussion section:

“[…], it is possible that expectation suppression is partially accounted for by an increased arousal in response to surprising stimuli. A related explanation is that enhanced pupil dilation to surprising stimuli (Damsma and van Rijn, 2017; Kloosterman et al., 2015; Preuschoff et al., 2011) results in enhanced retinal illumination, which in turn leads to stronger responses in early visual areas (Haynes et al., 2004), which could potentially also contribute to stimulus unspecific expectation suppression in V1. These interpretations are further supported by the fact that expectation suppression and pupil dilation differences between unexpected and expected attended stimuli were associated, with trailing images that elicit larger pupil dilation differences also showing more pronounced expectation suppression in V1.”

In addition, it could be informative to know whether the neural (and pupil) effects predict behavioral performance (although this may be difficult because behavioral performance was at ceiling in the attended condition). In any case, additional analyses linking pupil data to behavioral data and/or fMRI would be helpful for supporting the conclusions.

We agree with the reviewers that linking the neural effects of expectation with its behavioral benefits is useful. A particularly interesting question is whether the reaction time (RT) benefits due to valid expectations are linked to the neural expectation suppression effect. A prediction would be that stronger expectation suppression correlates with larger RT benefits due to valid expectations. Testing this prediction is challenging in the present dataset, given that expectation suppression is a difference between expectation conditions (i.e. not calculable on single trial level), thus precluding trial-wise analysis. However, it is possible to assess whether within subjects there is a correlation between the magnitude of expectation suppression and the expectation RT benefit per trailing image. We added the following analysis’s description to the Materials and methods subsection “Linking behavioral and neural measures”.

The corresponding results are presented in the manuscript’s Results subsection “Neural and behavioral effects of expectations are associated”., including Figure 4B.

The new results are briefly discussed in the last paragraph of the Discussion section:

“Finally, our results also demonstrate that larger expectation suppression effects in V1 and TOFC are associated with increased reaction time benefits afforded by expectations when people are judging the predictable objects. […] Predictions may thus help in converging more rapidly on an interpretation of the current sensory input, thereby contributing to faster reactions to expected than unexpected stimuli.”

2) Reviewers were surprised that no constriction (pupil light reflex) was found at stimulus onset, which presumably involves increases in contrast and light. Were the stimuli and ITI period matched in luminance? They for sure differ in contrast, to which the pupil also responds. Was the screen was quite dim in the scanner, so no transient constriction was elicited? Or was it perhaps that due to the quite narrow ITI range trial onset was quite predictable?

We agree with the reviewers that the absence of a pupil light response may initially seem surprising. However, as speculated by the reviewer, the stimuli and ITI period were indeed very similar in terms of their luminance. We think that this is important information for the reader and thus added the following paragraph to the “Stimuli” subsection of the Materials and methods: “We calculated the average relative luminance of the object stimuli by converting the stimulus images from sRGB to linear RGB, then calculated the relative luminance for all pixels (where relative luminance Y = 0.2126*R + 0.7152*G + 0.0722*B; Stokes et al., 1996), and finally averaged the obtained luminance values, thereby obtaining the mean relative luminance per image. On this relative luminance scale, 0 would be a completely black image, while 1 would be a white image. The average relative luminance of the stimulus set was 0.225, while the relative luminance of the mid gray background, presented during the ITI, was 0.216.” Thus, when considering the mean relative luminance of the stimuli compared to the ITI (background), one would not expect a reliable pupil light response.

Also, baseline pupil should be tested for the 2 tasks (e.g. the raw units) to see whether baseline arousal is different between conditions. This might help elucidate the phasic differences as well, since a lower baseline might be associated with a higher phasic response. Also, the increased dilation comes very late, when the pupil is returning to baseline, suggesting prolonged processing for unexpected stimuli. The baseline was taken from -1 s, giving the pupil response only 4 s to die out for the shortest ITI, which is quite short. The authors could control for expectedness in the previous trial to take into account the supposedly higher baseline after an unexpected trial.

We followed the reviewers suggestion and compared pupil size during the baseline periods of the two tasks and expectation conditions in raw units. The following subsection was added to the Supplemental analyses section (“No differences in pupil dilation during baseline”.

3) There are several points that need a more thorough discussion.a) While the authors frame their study as an attention study, one could also consider the finding an effect of task (task relevant vs. task irrelevant). Of course, the two concepts overlap, but this should be discussed.

We agree with the reviewer that we withdrew attention from the object stimuli by manipulating task-relevance, and as such, our findings, if considered in isolation, can be interpreted as an effect of attention or task-relevance. It is however by considering our results in the context of the relevant literature that we believe our results are better accounted for by a manipulation of attention than task-relevance. We fully agree that a discussion of the overlap and distinction between attention and task-relevance is crucial. Thus, we introduce the two concepts early in the manuscript (Introduction section): “On the other hand, den Ouden et al., 2009, demonstrated attenuated responses to task-irrelevant expected stimuli, suggesting the possibility that the sensory consequences of statistical learning may not depend on attention. et al.[…] Without competition, it is likely that even a task-irrelevant stimulus will receive some attention”.

We later repeat this argument in the Discussion, and now also explicitly point out the overlap between task-relevance and attention with respect to our design. “However, in all these studies, while the predictable stimuli were task-irrelevant, attention was not effectively drawn away by a competing stimulus that required attention. […] This is a crucial difference between the present and previous studies, because it is likely that any supraliminal stimulus, in the absence of competition, will be attended to some degree, even if it is not task-relevant, especially if the stimulus is surprising (Horstmann et al., 2015)”. Thus, previous studies have shown that task-irrelevant stimuli can result in a modulation of sensory responses by expectations (den Ouden et al., 2009; Kok et al., 2012a), while in our manipulation expectation effects vanish. Therefore, we believe that attention, rather than task-relevance, is likely to be the key manipulation that abolishes expectation effects.

Additionally, we show that task relevance of the predictive relationship itself does not seem to modulate expectation suppression either. Thus, synthesizing our results with those of previous studies, we conclude “that expectation suppression in the visual system occurs irrespective of exact task goals and relevance of the predictable objects and their predictable relationship, but it is abolished by drawing attention away from the stimuli”.

b) Is the explanation for the discrepancy with other studies related to differently controlled attention (as suggested by the authors) or, possibly, to differences in the nature of the stimuli and differences in how they are processed? Importantly, the statistical learning in this study was not automatic (even if being engaged in a different task participants were informed about the regularities) and thus it is likely that they paid attention to the regularities (the other task was easy – performance at ceiling), which might be a crucial difference to other settings. For example, there are auditory, but also visual, mismatch negativity studies which show mismatch effects when the stimuli are not attended. It seems highly unlikely that all of these studies did not control correctly for attention.

We think that the reviewer addresses an important point when referring to studies which show mismatch effects when stimuli were not attended. Indeed, we do believe that there are cases in which prediction (mismatch) effects can occur in the absence of (top-down) attention, as for example in classical mismatch negativity studies. However, it is important to keep in mind that these studies usually do not differentiate between expectation and adaptation (repetition), and thus do not allow for inferences with respect to the automaticity of either mechanism, just that at least one can induce a mismatch effect without attention. Indeed, there is evidence that adaptation (repetition) effects can persist with minimal attention (Chee and Tan, 2007; Murray and Wojciulik, 2004) or are unmodulated by attention (Bentley et al., 2003; although also different results have been reported, e.g. Eger et al., 2004). Thus, it is difficult to compare our results to studies that did not independently manipulate or control expectation and adaptation. However, with respect to studies that did investigate expectation effects while controlling for stimulus repetition, such as den Ouden et al., 2009, or Kok et al., 2012a; 2012b, a crucial distinction is the attention compared to a task-relevance manipulation. As explained above, and in our Discussion: “However, in all these studies, while the predictable stimuli were task-irrelevant, attention was not effectively drawn away by a competing stimulus that required attention. […] This is a crucial difference between the present and previous studies, because it is likely that any supraliminal stimulus, in the absence of competition, will be attended to some degree, even if it is not task-relevant, especially if the stimulus is surprising (Horstmann et al., 2015)”.

That said, we do agree with the reviewer that this should not be taken to suggest that all expectation effects depend on attention, but that is important to clarify the type of expectation that is being investigated. As we note in our Discussion: “[…] other, more ‘stubborn’ prior expectations (Yon et al., 2019) that are derived over longer (ontogenetic or phylogenetic) time scales may persist even when attention is drawn away, such as perceptual fill-in during the Kanizsa illusion (Kok et al., 2016)”. Indeed, classical mismatch negativity studies may depend on such more fundamental priors and the above mentioned repetition effects. The other suggestion made by the reviewer that differences in stimuli, and how they are processed, could account for differences in results between the present and previous studies, is important to consider in this context as well. It is indeed conceivable that for complex associations and stimuli (such as the object stimuli used here) the resolution of the expectation status may depend more on recurrent processing across the ventral visual stream, than for simpler associations and stimuli (e.g. grating stimuli, or dot and sound sequences). Expectations of a simpler type (simple associations and stimuli) may be resolved mainly within lower sensory areas. If this is the case, it does also seem plausible to suggest that the automaticity of expectation suppression may differ between these scenarios.

We now also discuss this possibility in the manuscript: “Similarly, for simple stimuli, such as oriented gratings (Kok et al., 2012a; Kok et al., 2012b) or simple sequences (Ekman et al., 2017), the resolution of expectations may depend less on recurrent processing throughout the visual hierarchy than for complex objects. Thus, it is conceivable that the automaticity of predictive processing partially depends on the complexity of the predictable stimuli and their association, with increasing complexity requiring increasing processing across the hierarchy, and in turn a focus of attention on the predictable stimuli.”.

Finally, it should be noted that while the task performance was indeed at ceiling during MRI scanning, participants frequently reported not noticing any associations (not even during behavioral training), and consequently also stated that they did not pay attention to the regularities during MRI scanning. This is also reflect by the fact that performance during the post-scanning recognition test (Figure 5B) is far from perfect. Combined this data suggest that participants were less aware of the associations than one may expect given the ceiling performance during MRI scanning, which in turn implies that the present study may not differ substantially in terms of the automaticity of the statistical learning from other studies of statistical learning.

c) The implications of the current findings for models of predictive coding need a more balanced discussion. While reviewers agree that the current evidence is not strongly supportive of the idea that attention provides gain control on prediction errors, their main argument in support of this may not be fully correct: It is well conceivable that there is very strong gain control on prediction error neurons (possibly a small subset of all neurons) even if there is strong baseline activity in response to all stimuli. Why should gain control affect all neurons in a region and not be neuron specific?

We agree with the reviewer that our discussion could have been better balanced. We reexamined the discussion of our results in the context of predictive coding, and have revised the corresponding subsection, “Attention and prediction errors”.

References:

Bentley, P., Vuilleumier, P., Thiel, C. M., Driver, J., & Dolan, R. J. (2003). Effects of attention and emotion on repetition priming and their modulation by cholinergic enhancement. Journal of Neurophysiology, 90(2), 1171–1181.

Chee, M. W. L., & Tan, J. C. (2007). Inter-relationships between attention, activation, fMR adaptation and longterm memory. NeuroImage, 37(4), 1487–1495.

Eger, E., Henson, R.N.A, Driver, J., & Dolan, R.J. (2004). BOLD Repetition Decreases in Object-Responsive Ventral Visual Areas Depend on Spatial Attention. J Neurophysiol. 92(2), 1241-7

Murray, S.O. & Wojciulik, E. (2004). Attention increases neural selectivity in the human lateral occipital complex. Nat. Neurosci., 7, 70–74.